# One Arrow, Two Hawks: Sharpness-aware Minimization for Federated Learning via Global Model Trajectory

**Yuhang Li** [1]   **Tong Liu** [1]   **Yangguang Cui** [1]   **Ming Hu** [2]   **Xiaoqiang Li** [1]

## Abstract

Federated learning (FL) presents a promising strategy for distributed and privacy-preserving learning, yet struggles with performance issues in the presence of heterogeneous data distributions. Recently, a series of works based on sharpness-aware minimization (SAM) have emerged to improve local learning generality, proving to be effective in mitigating data heterogeneity effects. However, most SAM-based methods do not directly consider the global objective and require two backward pass per iteration, resulting in diminished effectiveness. To overcome these two bottlenecks, we leverage the global model trajectory to directly measure sharpness for the global objective, requiring only a single backward pass. We further propose a novel and general algorithm `FedGMT` to overcome data heterogeneity and the pitfalls of previous SAM-based methods. We analyze the convergence of `FedGMT` and conduct extensive experiments on visual and text datasets in a variety of scenarios, demonstrating that `FedGMT` achieves competitive accuracy with state-of-the-art FL methods while minimizing computation and communication overhead. Code is available at https://github.com/harrylee999/FL-SAM.

## 1. Introduction

Federated Learning (FL) leverages distributed client (i.e., edge device) data to preserve privacy through an iterative process: downloading models, training models locally, uploading the updated models by clients and aggregating models on the server (McMahan et al., 2017). Due to the limited communication resources, only a subset of clients partici-

pate in the FL process and train the local models in multiple intervals with their own datasets within one communication round (Wahab et al., 2021). Due to data heterogeneity (i.e., non-IID), partial client participation, and multiple local training, a significant issue arises known as the "client drift" problem (Karimireddy et al., 2020), which means clients' local objectives converging towards inconsistent local optima, which leads to the aggregated global model may not be the optimum of the global objective. Larger data heterogeneity may enlarge the objective inconsistency, thereby not only degrading generalization performance but also escalating computation and communication overhead.

To tackle the client drift problem, most previous works (Li et al., 2020; Karimireddy et al., 2020; Wang et al., 2020b; Acar et al., 2021) involve enforcing regularization in local optimization with Empirical Risk Minimization (ERM). However, ERM-based training often falls into a sharp valley in the loss landscape (Chaudhari et al., 2019), resulting in a biased regularization term (e.g., the global model), especially in a highly heterogeneous dataset (Sun et al., 2023a). This bias makes the trained local model more inclined towards a similar tendency, rendering the entire training process unstable and significantly undermining performance. Instead, recent studies (Qu et al., 2022; Caldarola et al., 2022; Sun et al., 2023b; An et al., 2023; Dai et al., 2023) have focused on improving the local generality in FL to alleviate the objective inconsistency, which has achieved new state-of-the-art performance among current FL algorithms. These methods aim to seek flat minima during local learning by employing the recently proposed Sharpness-Aware Minimization (SAM) (Foret et al., 2020) as the local optimizer. SAM is built on the findings that emphasize the advantages of a smooth loss landscape for better generalization (Dinh et al., 2017; Jiang et al., 2019). By enhancing local generality, these methods mitigate inconsistency among local objectives, thereby contributing to the overall smoothness of the aggregated global model.

Unfortunately, SAM-based methods have following shortcomings. ① The computational cost of SAM's sharpness measure is twice that of the base optimizer, typically stochastic gradient descent (SGD). In the process of FL, most of the computation is performed by client devices. This prohibits

---

[1]School of Computer Engineering and Science, Shanghai University, Shanghai, China [2]School of Computing and Information Systems, Singapore Management University, Singapore. Correspondence to: Tong Liu <tong_liu@shu.edu.cn>.

*Proceedings of the 42^{nd} International Conference on Machine Learning*, Vancouver, Canada. PMLR 267, 2025. Copyright 2025 by the author(s).

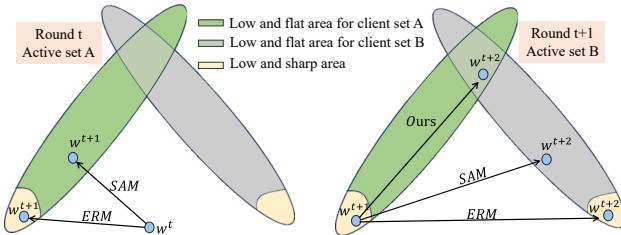

*Figure 1.* Due to partial participation in FL, client set A is active in round $t$ and set B in round $t + 1$. In round $t$, ERM training falls into a low and sharp area (cream), while SAM falls into a low and flat area (green) for the active client set A. In round $t + 1$, both ERM and SAM fall into a low area for the active client set B, but they may forget the knowledge learned from set A in the previous round. Our method, however, guides the global model toward a joint low and flat area (green+grey) for both sets A and B.

SAM from being deployed extensively in practical scenarios since most clients may have limited computing power and tend to minimize computing energy consumption (Li et al., 2023). ② Moreover, the sharpness measure of SAM on each client can only compute local model's sharpness direction to minimize. Thus minimizing local sharpness can not directly optimize the flatness of the global model, which may deviate from the global objective and slow down convergence, as shown in Figure 1.

To overcome data heterogeneity and the pitfalls of previous SAM-based methods, we propose a novel and general algorithm, FedGMT (**Fed**erated learning with **G**lobal **M**odel **T**rajectory), to guide each client search for a consistent smooth loss landscape aligned with global objective, thereby significantly improving overall performance. Specifically, we introduce a novel global model trajectory loss to directly measure and optimize the sharpness of the global model. This loss replaces SAM's sharpness measure loss by measuring the Kullback–Leibler (KL) divergence between the outputs of neural networks with the current local model and those with the past global models, which can guarantee each local update avoids falling into a sharp valley of the loss landscape from a global perspective. Furthermore, to prevent our global sharpness measure from being affected by data heterogeneity, we introduce a constraint to the global objective and use the alternating direction method of multipliers (ADMM) (Boyd et al., 2011) to solve. This ensures an accurate update direction to smooth the global model.

In the end, we summarize our main contributions as follows:

- We take a closer look at sharpness-aware minimization in heterogeneous FL from a global view. We propose a novel global model trajectory loss to directly measure the sharpness of the global model without information leakage of local data, supported by theoretical analysis. Furthermore, our measure's computation cost is nearly about $0.67\times$ of FedSAM's (Qu et al., 2022; Caldarola

et al., 2022) in each local update.

- To directly reduce the global model's sharpness and effectively address data heterogeneity, we propose FedGMT, which achieves a fast convergence speed with less computation cost and maintains high generalization. Theoretically, we provide the convergence rate upper bound under the non-convex and smooth cases and prove that FedGMT could achieve a fast convergence rate of $\mathcal{O}(1/T)$.

- We empirically show the effectiveness of FedGMT, which outperforms several SOTA baselines. FedGMT is also robust to various data heterogeneity levels, client participation levels and model architecture on distributed visual and text datasets.

## 2. A Closer Look at SAM-based Algorithms

In this section, we first formally describe the problem setup for FL, review SAM-based methods, and then outline our motivation. A detailed related work is in Appendix B.

### 2.1. Preliminaries

**Federated Learning (FL).** In a classic FL setting with $M$ clients, where each client has a dataset $\mathcal{D}_m$, the optimization problem to solve can be formulated as follows:

$$\min_w \left\{ \mathcal{L}(w) = \frac{1}{M} \sum_{m \in \mathbb{M}} \mathcal{L}_m(w) \right\}, \quad (1)$$
$$\mathcal{L}_m(w) \triangleq \mathbb{E}_{\xi_m \sim D_m} \ell(f(w; \xi_m)),$$

where $w$ is the model parameter, $\mathbb{M}$ is the set of all clients, $\mathcal{L}_m(w)$ is the empirical risk minimization (ERM) for client $m$, $\ell$ is a loss function (e.g. cross-entropy loss), $f$ denotes a neural network and $\xi_m$ denotes the pair (inputs, targets) of a randomly sampled instance from $\mathcal{D}_m$. At each round $t$, FedAvg (McMahan et al., 2017) formulates this minimization problem as performing a weighted average of the local model parameters $w_m^t$ updated by the subset of selected clients $\mathbb{N}^t$. Reddi et al. (2021) shows that the FedAvg global update can be generally seen as one step of SGD with a unitary learning rate:

$$w^{t+1} = \frac{1}{N} \sum_{m \in \mathbb{N}^t} w_m^t = w^t - \eta_g \frac{1}{N} \sum_{m \in \mathbb{N}^t} (w^t - w_m^t), \quad (2)$$

where $\eta_g$ is the server-side learning rate, equal to 1 in FedAvg. The difference $w^{t+1} - w^t := \Delta^t$ defines the global pseudo-gradient at round $t$.

**Sharpness-Aware Minimization (SAM).** SAM (Foret et al., 2020) focuses on optimizing the sharp points from parameter space so that the training model can produce a flat

*Table 1.* Overview of several SAM-based algorithms in FL. The communication cost is defined as the parameters transmitted per round (1.5× for `FedGMT` and 1× for `FedGMTv2`). The computational cost is defined as per-iteration training cost (1.33× is based on the assumption that backward cost of is twice forward). In `FedSMOO`, $\mu_m$ and $s$ are dual variable and correction to perturbations. In `FedLESAM`, $w^{old}$ is the global model received at previous active round. Other symbols are detailed in Table A in Appendix.

| Research work | Global Flatness | Sharpness Measure | Communication Cost | Computation Cost |
|---|---|---|---|---|
| `FedSAM` (ECCV22, ICML22) | × | $\mathcal{L}_m(w_m^t + \rho \frac{\nabla \mathcal{L}_m(w_m^t)}{\|\nabla \mathcal{L}_m(w_m^t)\|}) - \mathcal{L}_m(w_m^t)$ | 1× | 2× |
| `FedSpeed` (ICLR23) | × | Similar to `FedSAM` | 1× | 2× |
| `FedSMOO` (ICML23) | ✓ | $\mathcal{L}_m(w_m^t + \rho \frac{\nabla \mathcal{L}_m(w_m^t) - \mu_m - s}{\|\nabla \mathcal{L}_m(w_m^t) - \mu_m - s\|}) - \mathcal{L}_m(w_m^t)$ | 2× | 2× |
| `FedLESAM` (ICML24) | ✓ | $\mathcal{L}_m(w_m^t + \rho \frac{w^{old} - w^t}{\|w^{old} - w^t\|}) - \mathcal{L}_m(w_m^t)$ | 1× | 1× |
| `FedGMT` (Ours) | ✓ | $\ell_{KL}(f(e^t), f(w_m^t))$ | 1 × or 1.5× | 1.33× |

loss landscape. To achieve this, SAM applies a weight perturbation vector $\epsilon$ to the model parameters $w$ and conducts the following objective:

$$\mathcal{L}^{SAM}(w) \triangleq \mathcal{L}(w + \epsilon), \text{ where } \epsilon \triangleq \rho \frac{\nabla \mathcal{L}(w)}{\|\nabla \mathcal{L}(w)\|}. \quad (3)$$

Here, $\|\cdot\|$ is $l_2$-norm, $\rho$ is the radius of the neighborhood and $\nabla$ is the abbreviation for $\nabla_w$ on parameters $w$.

While Eq. (3) improves performance, computing $\epsilon$ and $\mathcal{L}(w+\epsilon)$ adds an extra backward and forward pass, doubling SAM's runtime compared to SGD. Minimizing Eq. (3) can also be interpreted as jointly minimizing the loss value and the sharpness of the loss landscape defined by:

$$\textbf{Sharpness}: S(w) \triangleq \mathcal{L}(w + \epsilon) - \mathcal{L}(w). \quad (4)$$

### 2.2. Rethinking SAM-based Algorithms

We first summarizes several SAM-based methods in FL at Table 1. `FedSAM` (Qu et al., 2022; Caldarola et al., 2022) directly applies the SAM objective to replace the local ERM objective, which can be formulated as a change in Eq. (1): $\mathcal{L}_m(w) \rightarrow \mathcal{L}_m(w + \epsilon_m)$, where $\epsilon_m$ is the local weight perturbation allocated to $\mathcal{L}_m$. Notably, the perturbation $\epsilon_m$ is computed based on the local dataset $\mathcal{D}_m$. Consequently, the utilization of this local SAM optimizer yields effective generalization only on datasets drawn independently and identically as $\mathcal{D}_m$. Intuitively, enhancing the generality of local training individually on $\mathcal{D}_m$ would inherently improve generalization on global dataset $\mathcal{D} = \cup_m \mathcal{D}_m$. However, this indirect promotion may not be effective when dealing with severely heterogeneous datasets. In Fig. 2, we empirically demonstrate that `FedSAM` minimizes sharpness slightly compared to `FedAvg` in severely heterogeneous scenarios. This observation also explains why `FedSAM` does not outperform `FedAvg` in this case. `FedSpeed` (Sun et al., 2023b) incorporates a local SAM optimizer with a dynamic regularizer to enhance its effectiveness. This approach aims to improve local consistency to bridge the smooth information on both local and global models.

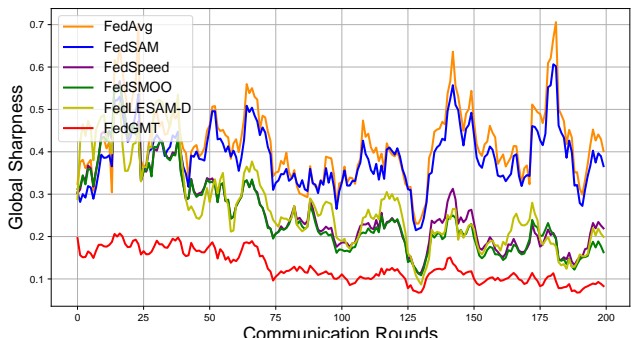

*Figure 2.* The sharpness change of the global model on CIFAR-10 in the high data heterogeneity scenario (Dir(0.01)).

Similar approaches like `MoFedSAM` (Qu et al., 2022) and `FedMRUR` (An et al., 2023) inject momentum into the local SAM optimizer and `FedGAMMA` (Dai et al., 2023) combines the `SCAFFOLD` (Karimireddy et al., 2020) with SAM to improve the performance. Although these methods contribute to minimizing sharpness to some extent, the flatness of the global landscape still can not be directly optimized. This is due to the fact that the perturbation $\epsilon_m$ is not designed for the global function $\mathcal{L}$.

To directly smooth the global model, it is necessary to modify $\mathcal{L}_m(w) \rightarrow \mathcal{L}_m(w + \epsilon)$, where $\epsilon$ is the global weight perturbation allocated to $\mathcal{L}$. However, $\epsilon$ needs to be calculated on the global dataset $\mathcal{D}$, which is not feasible in FL. `FedSMOO` (Sun et al., 2023a) utilizes the ADMM to estimate global perturbation $\epsilon$ and also adopts a dynamic regularizer during the local training. `FedLESAM` (Fan et al., 2024) estimates the global perturbation as the difference between the global model from the previous round and the global model received in the current round. However, these estimations are not significantly effective compared to `FedSpeed` which directly utilizes local SAM in Fig. 2.

**Motivation**: Given that SAM aims to yield a flat loss landscape by minimizing sharpness during training, we utilize a first-order Taylor expansion to decompose the sharpness

measure of SAM in Eq. (4) for the global function $\mathcal{L}$ in Eq. (1) at round $t$ as:

$$
\begin{aligned}
S_{\mathbb{N}^t}(w^t) &= \mathcal{L}_{\mathbb{N}^t}(w^t + \epsilon) - \mathcal{L}_{\mathbb{N}^t}(w^t) \\
&\approx \mathcal{L}_{\mathbb{N}^t}(w^t) + \epsilon \Delta_{\mathbb{N}^t}^t - \mathcal{L}_{\mathbb{N}^t}(w^t) \\
&\approx \rho \cdot \Delta_{\mathbb{N}^t}^{t\ \top} \Delta_{\mathbb{N}^t}^t / \|\Delta_{\mathbb{N}^t}^t\| \\
&= \rho \|\Delta_{\mathbb{N}^t}^t\|.
\end{aligned}
\tag{5}
$$

From Eq. (5), we observe that the $l2$-norm of the global pseudo-gradient $\|\Delta_{\mathbb{N}^t}^t\|$ could be approximated as the global sharpness measure in FedLESAM according to the analysis in (Zhao et al., 2022). However, in practical FL, due to data heterogeneity and the high probability of varying active clients $\mathbb{N}^t$ between consecutive rounds, the global pseudo-gradients in two consecutive rounds differ. Thus a single previous global pseudo-gradient may not guarantee the right direction toward global flatness in the current round.

To address this, as a single global pseudo-gradient is ineffective and unreliable, our approach use the global model trajectory which includes the pseudo-gradients from all the previous rounds. This significantly reduces sharpness compared to other SAM-based algorithms, as shown in Fig. 2.

## 3. Proposed Method: FedGMT

In this section, we introduce FedGMT and FedGMT-v2, an enhanced variant designed to save communication overhead. The detailed algorithm is outlined in Algorithm 1.

### 3.1. Measure Sharpness via Global Model Trajectory

We denote the past trajectory of the global model as set $\Omega = \{w^0, w^1, \ldots, w^{t-1}\}$. To avoid excessive storage and effectively utilize all the information in $\Omega$, we adopt an exponential moving average (EMA) weighting strategy on the global model's update trajectory. The EMA model $e^t$ at the $t$-round is updated as follows:

$$
e^t = \alpha e^{t-1} + (1 - \alpha) w^t,
\tag{6}
$$

where $\alpha \in (0, 1)$ is the coefficient of EMA. On the other hand, given that $e^0 = w^0$ and $w^{t+1} = w^t - \Delta_{\mathbb{N}^t}^t$ then the relation between the EMA model $e^t$ and the global model $w^t$ as follows:

$$
e^t = w^t + \sum_{i=0}^{t-1} \alpha^{t-i} \Delta_{\mathbb{N}^i}^i.
\tag{7}
$$

More details of this derivation can be found in Appendix C. Therefore, the EMA model $e^t$ includes the pseudo-gradients from all the previous rounds and puts more emphasis on the more recent pseudo-gradients.

To connect the EMA model $e^t$ to sharpness minimization, inspired by (Du et al., 2022) which uses loss difference, we consider the trajectory from $e^t$ to $w^t$ as $\Theta =$

$\{\theta^0, \theta^1, \ldots, \theta^{t-1}\}$, where $\theta^i = \theta^{i-1} - \alpha^{t-i} \Delta_{\mathbb{N}^i}^i$, $\theta^0 = e^t$ and $\theta^t = w^t$. The sharpness measure $S_{\mathbb{N}^t}(w^t)$ in Eq. (4) is non-negative, and thus we have:

$$
\begin{aligned}
\arg\min_{w^t} S_{\mathbb{N}^t}(w^t) &= \arg\min_{\theta^t} S_{\mathbb{N}^t}(\theta^t) \\
&\approx \arg\min_{\theta^t} [\mathcal{L}_{\mathbb{N}^t}(\theta^0) - \mathcal{L}_{\mathbb{N}^t}(\theta^t)] \\
&= \arg\min_{w^t} [\mathcal{L}_{\mathbb{N}^t}(e^t) - \mathcal{L}_{\mathbb{N}^t}(w^t)].
\end{aligned}
\tag{8}
$$

More details of this derivation can be found in Appendix C.2. Therefore, minimizing $\mathcal{L}_{\mathbb{N}^t}(e^t) - \mathcal{L}_{\mathbb{N}^t}(w^t)$ is equivalent to minimizing the SAM's sharpness measure $S_{\mathbb{N}^t}(w^t)$ for the global function $\mathcal{L}$.

Based on the above result, if we directly combine the global trajectory loss term $\mathcal{L}(e^t) - \mathcal{L}(w^t)$ in Eq. (8) with FL objective $\mathcal{L}(w^t)$ in Eq. (1), the $\mathcal{L}(w^t)$ will unfortunately be canceled out. Without loss of generality, we replace the cross entropy loss with the KL divergence loss to decouple the vanilla loss. Then the loss function $\mathscr{L}_m$ of FedGMT for the $m$-th client at $t$ round as:

$$
\mathscr{L}_m(w_m^t, e^t) := \overbrace{\mathcal{L}_m(w_m^t)}^{ERM} + \overbrace{\mathcal{L}_m^{glotra}(w_m^t, e^t)}^{measure\ global\ sharpness}
$$
$$
\mathcal{L}_m^{glotra}(w_m^t, e^t) := \frac{\gamma}{|\mathcal{D}_m|} \sum_{\mathcal{D}_m} \ell_{KL}(f(e^t), f(w_m^t)),
\tag{9}
$$

where the hyperparameter $\gamma$ stands for the strength of minimizing sharpness.

**Remark 3.1.** Unlike the SAM, $\mathcal{L}_m^{glotra}$ in Eq. (9) only requires one more forward pass and no extra backward pass.

### 3.2. Promoting Global Consistency with ADMM

Note that an unbiased procedure to minimize global sharpness should use the global model $w^t$ and the EMA model $e^t$, i.e., to minimize $\mathcal{L}^{glotra}(w^t, e^t)$ on the global dataset $\mathcal{D}$. However, in FL framework, the global model $w^t$ is sent to each client $m$ and split into $w_m^t$ for independent updates as Eq. (9). This process introduces non-vanishing biases $w_m^t - w^t$ to the global sharpness measure, potentially diminishing algorithm performance.

To bridge this gap, the local update should align with the global update. Therefore, we takes the form of an ADMM-like method in order to align client and server and effectively minimize the global objective $\mathcal{L}$. ADMM makes use of the augmented Lagrangian function via penalizing the constrain $w_m^t = w^t$. We first scale the KL function to a non-negative and convex function as:

$$
\mathcal{L}_m^{glotra}(w_m^t) \le \frac{1}{2\beta} \left\| w^t - w_m^t + \sum_{i=0}^{t-1} \alpha^{t-i} \Delta^i \right\|^2,
\tag{10}
$$

where $\beta$ is a penalty coefficient. More details of this derivation can be found in Appendix D.

By using the first-order derivative of Eq. (10), we define the global augmented Lagrangian function as:

$$\frac{1}{M}\sum_m \mathscr{F}_m(w_m) + u_m(w - w_m) + \frac{1}{2\beta}\|w - w_m\|^2, \quad (11)$$

where $\mathscr{F}_m(w_m^t) := \mathcal{L}_m(w_m^t) - \langle\frac{1}{\beta}\sum_{i=0}^{t-1}\alpha^{t-i}\Delta^i, w_m^t\rangle$ and $u_m$ is the dual variable. We denote $\langle\cdot,\cdot\rangle$ as the inner product for two vectors.

To solve the minimization, we decompose the problem in Eq. (11) for each client to solve as :

$$\begin{aligned} w_{m,K}^t &= \arg\min_{w_m^t} \mathscr{F}_m(w_m^t) - \langle u_m^t, w_m^t\rangle + \frac{1}{2\beta}\|w^t - w_m^t\|^2 \\ &= \arg\min_{w_m^t} \mathcal{L}_m(w_m^t) - \langle u_m^t, w_m^t\rangle, \quad (12) \end{aligned}$$

where $w_{m,K}^t$ is the local model after K local interval updates. Then we update the dual variable as $u_m^{t+1} = u_m^t - \frac{1}{\beta}(w_m^t - w^t)$. Lastly, by using the first-order derivative of Eq. (11), the global model $w$ with partial participation is updated as:

$$\begin{aligned} w^{t+1} &= \arg\min_{w^t} \frac{1}{N}\sum_{m\in\mathbb{N}^t}\|w_{m,K}^t - w^t - \beta u_m^{t+1}\|^2 \\ &= \frac{1}{N}\sum_{m\in\mathbb{N}^t}(w_{m,K}^t - \beta u_m^{t+1}). \quad (13) \end{aligned}$$

To avoid each client sending $u_m^{t+1}$ to the server in each round, we define the global dual variable $u^{t+1} := u^t - \frac{1}{\beta M}\sum_{m\in\mathbb{N}^t}(w_{m,K}^t - w^t)$. Then, we reformat Eq. (13) as $w^{t+1} = \frac{1}{N}\sum_{m\in\mathbb{N}^t} w_{m,K}^t - \beta u^{t+1}$.

With these update rules, we can calculate the local models of approximately satisfying the constraints. Intuitively, with the consistent local updates, minimizing the global trajectory loss $\mathcal{L}_m^{glotra}$ among clients would mitigate the impact of significant changes in training sample loss within the global loss function. This helps prevent overfitting to local datasets by enforcing their predictions to be close to the ones from the EMA model. Therefore, each client can search for a consistent smooth loss landscape aligned with the global objective, thereby significantly improving overall performance.

### 3.3. Enhanced Variant for Communication-Efficient

The whole workflow of FedGMT is shown in Algorithm 1. At the beginning of each round $t$, we randomly select a subset of active clients $\mathbb{N}^t$ from the total clients set $\mathbb{M}$. The global server will communicate the parameters $w^t$ and $e^t$ to the active clients for local training. Since the communication with the server for $e^t$ increases communication cost by doubling the message size in each round, inspired by SCAFFOLD and FedLESAM, clients can utilize the EMA

**Algorithm 1** FedGMT and FedGMT-v2 Algorithm

**Input:** Model parameters $w$, EMA model parameters $e$, communication round $T$, local interval $K$, dual variable $u$, EMA coefficient $\alpha$, penalty coefficient $\beta$, learning rate $\eta$.

**Output:** Global model parameters $w^T$.
1: **Initialization** : $w = e = w^0, u_m = u = 0$;
2: **for** $t = 0, 1, 2, \cdots, T - 1$ **do**
3:     randomly select the active clients set $\mathbb{N}^t$ from $\mathbb{M}$;
4:     send the $w^t$ and $e^t$ to the active clients;
5:     **for** client $m \in \mathbb{N}^t$ **in parallel do**
6:         Initialize local model as $w_{m,0}^t = w^t$;
7:         $e^t = \alpha e^t + (1 - \alpha)w^t$;
8:         **for** $k = 0, 1, \cdots, K - 1$ **do**
9:             sample a minibatch and do
10:             $g_{m,k}^t = \nabla\mathscr{L}_m(w_{m,k}^t, e^t)$;     ▷ Using Eq. (9)
11:             $w_{m,k+1}^t = w_{m,k}^t - \eta(g_{m,k}^t - u_m^t)$;
12:         **end for**
13:         $u_m^{t+1} = u_m^t - \frac{1}{\beta}(w_{m,K}^t - w^t)$;
14:         send the $w_m^t = w_{m,K}^t$ to the global server;
15:     **end for**
16:     $u^{t+1} = u^t - \frac{1}{\beta M}\sum_{m\in\mathbb{N}^t}(w_m^t - w^t)$;
17:     $w^{t+1} = \frac{1}{N}\sum_{m\in\mathbb{N}^t} w_m^t - \beta u^{t+1}$;
18:     $e^{t+1} = \alpha e^t + (1 - \alpha)w^{t+1}$;
19: **end for**

model in the previous active round. Thus we propose the variant FedGMT-v2. The calculation of $e^t$ in FedGMT-v2 is an approximation of that in FedGMT, which inevitably introduces a slight performance degradation but reduces the communication cost by half for real-world practicality.

### 3.4. Convergence Analysis

In this section, we will demonstrate the theoretical analysis of our proposed FedGMT and illustrate the convergence guarantees under the specific hyperparameters. Firstly, some common assumptions are stated as follows.

**Assumption 3.2.** ($L$-smooth). Function $\mathcal{L}_m(w)$ is $L$-smooth and neural network $f_m(w)$ is $L_f$-smooth for all $m \in \mathbb{M}$ and $w \in \mathbb{R}^d$.

**Assumption 3.3.** (Unbiased and bounded gradient). The stochastic gradients on a batch of client $m$'s data $\varepsilon_m$ is an unbiased estimator of $\nabla\mathcal{L}_m(w)$ with an upper bound $G$, i.e., $\mathbb{E}[\nabla\mathcal{L}_m(w, \varepsilon_m)] = \nabla\mathcal{L}_m(w)$ and $|[\nabla\mathcal{L}_m(w)]_j| \leq G$ for all $m \in \mathbb{M}$, $w \in \mathbb{R}^d$ and $j \in [d]$.

**Assumption 3.4.** Neural network $f_m$ outputs a non-negative probabilistic vector, e.g. the last layer is softmax, then there exists $\delta > 0$ such that $\min_{c\in[C]}[f_m(w, x)]_c \geq \delta > 0$ for all $m \in \mathbb{M}$, $x \in \mathbb{R}^{d'}$ and $w \in \mathbb{R}^d$. Here, $c$ is the class index.

Assumption 3.2 assumes the gradient Lipschitz continuity for the objective function and neural network. Assumption 3.3 assumes the bounded stochastic properties of the gradients. The above two assumptions are commonly used in the analysis of the FL framework (Luo et al., 2022; Chen et al., 2023). Assumption 3.4 assumes a lower bound for the neural network's output, which is utilized to scale the KL function. This assumption is also employed in (Yao et al., 2023; Yang et al., 2023) for convergence analysis.

Our theoretical analysis depends on the above assumptions to study the properties of the proposed method. Proof details could be referred to the Appendix D.

**Theorem 3.5.** *Let the above assumptions hold, when $\alpha \leq \frac{1}{\sqrt{6NT}}$, $\beta \leq \frac{\sqrt{2N}}{\sqrt{135+5\sqrt{5}}ML}$ and $\gamma \geq \frac{\sqrt{135+5\sqrt{5}}ML\delta}{2\sqrt{2}NL_f}$, the mean averaged parameters sequence $\left\{\overline{w}^{t+1} \triangleq \frac{1}{N}\sum_{m\in\mathbb{N}^t} w_m^t\right\}_{t\in[T-1]}$ generated by the Algorithm 1 under the non-convex case satisfy:*

$$\frac{1}{T}\sum_{t=1}^{T}\mathbb{E}\|\nabla\mathcal{L}(\overline{w}^t)\|^2$$
$$\leq \frac{1}{\kappa T}\left(\mathcal{L}(\overline{w}^1) - \mathcal{L}^* + \frac{20M\beta^2L^2}{N}\Phi^0 + \frac{18\beta G}{N}\right), \ (14)$$

*where $\Phi^0 = \frac{1}{M}\sum_{m\in\mathbb{M}}\mathbb{E}\|w_m^0 - \overline{w}^1\|^2$ is the inconsistent term at the first round for $\overline{w}^1 \triangleq \frac{1}{N}\sum_{m\in\mathbb{N}^t} w_m^0$, $\kappa$ is a positive constant, $\mathcal{L}^*$ is the optima of $\mathcal{L}$.*

**Remark 3.6.** Under a constant number of clients uniformly selected at random in each round, and suitable values of $\alpha, \beta$, and $\gamma$ are chosen, Algorithm 1 attains a fast convergence rate of $\mathcal{O}(1/T)$ with non-convex local losses, which matches the conclusion of existing works (Acar et al., 2021; Gong et al., 2022; Sun et al., 2023a;b). Moreover, the last two terms in Eq. (14) achieve $N\times$ linear speedup.

# 4. Experiments

## 4.1. Experimental Setup

**Datasets and Models.** We conduct extensive experiments containing computer vision (CV) and natural language processing (NLP) domains. For the CV domain, we consider the image classification tasks with three widely used datasets including CIFAR-10/100 (Krizhevsky Alex, 2009) and CINIC-10 (Darlow et al., 2018) using the CNN from (McMahan et al., 2017), ResNet-8 (He et al., 2016) and ViT (Dosovitskiy et al., 2021), respectively. For the NLP domain, we study the text classification tasks with AG News (Zhang et al., 2015) using FastText (Joulin et al., 2017).

**Heterogeneous Partition Strategy.** We consider two common data heterogeneity scenarios: Pathological (McMahan et al., 2017) and Dirichlet (Wang et al., 2020a) settings.

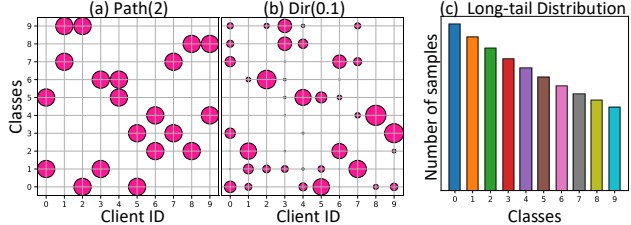

*Figure 3.* (a) & (b): Partition examples across 10 clients. (c): Long-tail distribution across category on CIFAR-10.

We represent the parameter $r$ as $Path(r)$ and $Dir(r)$ to control data heterogeneity, with smaller values of $r$ indicating increased data heterogeneity. Moreover, real-world data often exhibits a long-tail distribution characterized by significant class imbalance, and many SOTA FL methods have poor performance (Shang et al., 2022b) in such case. Therefore, to further enhance the data heterogeneity and simulate real-world scenarios, we follow (Cao et al., 2019; Shang et al., 2022b) to shape the original balanced dataset into a long-tail distribution for the Dirichlet setting. We set the imbalance factor (i.e., the ratio of instances in the most represented class to those in the least represented class) to 2 for all datasets. Figure 3 illustrates the differences in partition strategies and long-tail distribution.

**Baselines.** We take two lines of state-of-the-art methods as baselines. ① heterogeneity-oriented methods including FedAvg (McMahan et al., 2017), FedDyn (Acar et al., 2021) and FedNTD (Lee et al., 2022). ② SAM-based methods including FedSAM (Qu et al., 2022; Caldarola et al., 2022), FedSpeed (Sun et al., 2023b), FedSMOO (Sun et al., 2023a) and FedLESAM-D (Fan et al., 2024) (FedLESAM with dynamic regularizer). The introductions and hyperparameters for each method in the Appendix E.

**Common Training Details.** We set the batch size to 50 for CIFAR-10 and AG News, 20 for CIFAR-100, and 100 for CINIC-10. The number of local epochs is set to 5, except for CIFAR-100, in which we use 2 to mitigate extreme overfitting. We employ SGD with a learning rate of 0.01, momentum of 0.9, and weight decay of 1e-5, except for AG News, in which we use a learning rate of 0.1. The total number of communication rounds is set to 500, except for CINIC-10 and AG News, in which we conduct 1000 rounds, ensuring sufficient rounds for performance saturation. Following (Caldarola et al., 2022; 2023), we report the final averaged test accuracy and standard deviation over the last 50 rounds for increased robustness and reliability.

**FedGMT Setting.** The hyperparameters of FedGMT remain consistent across datasets, model architectures, and scenarios. We set the KL temperature to 3 and $\gamma = 1$ in $\mathcal{L}^{glotra}$, with the EMA coefficient $\alpha$ of 0.95 in FedGMT and 0.5 in FedGMT-v2 and the penalty coefficient $\beta$ chosen from $\{10, 100\}$ for all experiments. The detailed hyperparameters selection and sensitivity analysis in Section 4.4.

*Table 2.* Average (standard deviation) test accuracy@1(%) comparison. $p$: the participation ratio (%). **R&T(%)**: communication **r**ound and computation **t**ime (minutes) to achieve a target accuracy in brackets. ×: the target accuracy is not available.

| | Non-IID Partition Strategy : Pathological ($Path$) | | | | | | | | | |
|---|---|---|---|---|---|---|---|---|---|---|
| | Heterogeneity | | | Partial Participation | | | Communication and Computation | | | |
| Method | CIFAR-10 (CNN) | | | CIFAR-100-$Path$(10) (ResNet) | | | CINIC-10 (ViT) | | AG News (FastText) | |
| | $Path$(10) | $Path$(4) | $Path$(2) | $p=20\%$ | $p=10\%$ | $p=5\%$ | $Path$(2) | R&T(41%) | $Path$(2) | R&T(82%) |
| FedAvg | $78.91_{(0.68)}$ | $75.24_{(2.18)}$ | $65.28_{(5.39)}$ | $41.33_{(0.91)}$ | $37.47_{(1.25)}$ | $29.60_{(2.18)}$ | $34.38_{(4.55)}$ | × | $71.81_{(11.31)}$ | × |
| FedDyn | $80.69_{(0.36)}$ | $78.99_{(1.55)}$ | $74.13_{(3.63)}$ | $46.94_{(1.10)}$ | $45.04_{(1.03)}$ | $41.68_{(1.33)}$ | $41.63_{(5.30)}$ | 929  88.87 | $82.16_{(8.37)}$ | 710  66.74 |
| FedNTD | $79.65_{(0.25)}$ | $78.25_{(0.63)}$ | $73.36_{(1.75)}$ | $42.53_{(0.48)}$ | $40.72_{(0.51)}$ | $36.59_{(0.88)}$ | $49.27_{(2.17)}$ | 491  53.19 | $77.60_{(8.22)}$ | × |
| FedSAM | $80.60_{(0.69)}$ | $75.34_{(2.23)}$ | $66.07_{(4.81)}$ | $41.03_{(0.71)}$ | $36.93_{(1.27)}$ | $28.97_{(2.26)}$ | $36.34_{(4.19)}$ | × | $71.26_{(10.69)}$ | × |
| FedSpeed | $83.49_{(0.47)}$ | $80.40_{(1.50)}$ | $76.74_{(1.49)}$ | $47.10_{(0.70)}$ | $45.44_{(1.03)}$ | $40.06_{(1.61)}$ | $42.98_{(3.50)}$ | 688  149.30 | $85.19_{(3.51)}$ | 487  73.86 |
| FedSMOO | $82.92_{(0.34)}$ | $80.51_{(1.19)}$ | $76.59_{(2.08)}$ | $47.48_{(0.87)}$ | $45.76_{(1.12)}$ | $40.81_{(1.61)}$ | $44.15_{(3.53)}$ | 692  167.69 | $84.84_{(3.12)}$ | 548  83.80 |
| FedLESAM-D | $81.48_{(0.22)}$ | $79.79_{(0.57)}$ | $74.99_{(2.55)}$ | $47.09_{(0.79)}$ | $44.96_{(0.92)}$ | $40.95_{(1.88)}$ | $40.96_{(5.37)}$ | × | $85.37_{(2.57)}$ | 690  77.86 |
| **FedGMT** | $\mathbf{83.64}_{(0.11)}$ | $\mathbf{82.79}_{(0.22)}$ | $\mathbf{78.04}_{(1.03)}$ | $\mathbf{49.76}_{(0.33)}$ | $\mathbf{48.19}_{(0.56)}$ | $\mathbf{46.11}_{(0.67)}$ | $\mathbf{52.67}_{(1.13)}$ | **392  40.38** | $\mathbf{89.39}_{(0.62)}$ | **329  33.72** |
| **FedGMT-V2** | $83.25_{(0.13)}$ | $82.44_{(0.28)}$ | $78.01_{(1.06)}$ | $49.54_{(0.37)}$ | $47.95_{(0.39)}$ | $45.83_{(0.73)}$ | $51.97_{(1.75)}$ | 393  40.61 | $88.93_{(0.81)}$ | 351  35.10 |

| | Non-IID Partition Strategy : Long-tail Dirichlet ($Dir$) | | | | | | | | | |
|---|---|---|---|---|---|---|---|---|---|---|
| Method | CIFAR-10 (CNN) | | | CIFAR-100-$Dir$(0.1) (ResNet) | | | CINIC-10 (ViT) | | AG News (FastText) | |
| | $Dir$(1.0) | $Dir$(0.1) | $Dir$(0.01) | $p=20\%$ | $p=10\%$ | $p=5\%$ | $Dir$(0.1) | R&T(44%) | $Dir$(0.1) | R&T(85%) |
| FedAvg | $75.73_{(0.85)}$ | $70.61_{(3.51)}$ | $61.94_{(4.93)}$ | $41.96_{(0.53)}$ | $39.34_{(1.00)}$ | $35.13_{(1.69)}$ | $39.77_{(4.13)}$ | × | $77.48_{(6.87)}$ | × |
| FedDyn | $78.15_{(0.33)}$ | $75.71_{(0.95)}$ | $70.52_{(2.32)}$ | $46.93_{(0.52)}$ | $44.81_{(0.64)}$ | $43.04_{(0.84)}$ | $44.01_{(2.05)}$ | 985  80.11 | $85.72_{(1.58)}$ | 745  54.39 |
| FedNTD | $76.60_{(0.39)}$ | $72.62_{(1.73)}$ | $66.96_{(2.45)}$ | $41.39_{(0.45)}$ | $39.74_{(0.46)}$ | $37.52_{(0.65)}$ | $48.65_{(2.28)}$ | 654  53.41 | $83.55_{(2.55)}$ | × |
| FedSAM | $78.31_{(0.95)}$ | $70.96_{(3.97)}$ | $61.05_{(4.85)}$ | $41.78_{(0.63)}$ | $38.95_{(0.89)}$ | $34.63_{(1.61)}$ | $39.27_{(3.84)}$ | × | $80.25_{(4.93)}$ | × |
| FedSpeed | $80.38_{(0.38)}$ | $77.51_{(0.97)}$ | $71.96_{(1.67)}$ | $48.13_{(0.53)}$ | $45.73_{(0.52)}$ | $43.38_{(0.78)}$ | $47.57_{(2.34)}$ | 760  137.05 | $85.88_{(2.05)}$ | 687  80.61 |
| FedSMOO | $80.22_{(0.39)}$ | $77.08_{(0.97)}$ | $72.11_{(1.79)}$ | $48.26_{(0.42)}$ | $45.69_{(0.75)}$ | $43.06_{(0.79)}$ | $49.07_{(2.29)}$ | 712  160.20 | $85.74_{(2.15)}$ | 767  100.22 |
| FedLESAM-D | $78.02_{(0.27)}$ | $76.11_{(0.83)}$ | $71.10_{(2.82)}$ | $46.97_{(0.63)}$ | $45.21_{(0.67)}$ | $42.39_{(1.04)}$ | $43.00_{(2.51)}$ | × | $85.68_{(1.62)}$ | 745  67.30 |
| **FedGMT** | $\mathbf{80.48}_{(0.18)}$ | $\mathbf{79.17}_{(0.49)}$ | $\mathbf{74.67}_{(0.77)}$ | $\mathbf{48.65}_{(0.22)}$ | $\mathbf{47.42}_{(0.28)}$ | $\mathbf{45.46}_{(0.37)}$ | $\mathbf{57.84}_{(0.77)}$ | **404  33.40** | $\mathbf{88.39}_{(0.67)}$ | **484  37.51** |
| **FedGMT-V2** | $80.00_{(0.18)}$ | $78.73_{(0.47)}$ | $74.11_{(1.32)}$ | $48.64_{(0.19)}$ | $46.95_{(0.25)}$ | $44.66_{(0.33)}$ | $57.61_{(1.00)}$ | 413  35.45 | $88.31_{(0.82)}$ | 485  38.07 |

## 4.2. Performance Evaluation

**Datasets and Model Architecture.** FedGMT consistently exhibits efficacy across diverse datasets and model architectures. In Table 2, the performance of many baselines varies significantly with different datasets and model architectures. Moreover, FedGMT-v2 incurs only a slight performance reduction compared to FedGMT across all settings.

**Heterogeneity.** FedGMT is less affected by the degree of data heterogeneity. In Table 2, most baselines exhibit better performance in less heterogeneous settings (i.e., $path$(10) and $Dir$(1.0)). However, as data heterogeneity increases (i.e., $path$(2) and $Dir$(0.01)), they perform worse than FedGMT and FedGMT-v2.

**Partial Participation.** FedGMT demonstrates robustness to various levels of partial participation. As the participation ratio decreases, the data amount for training in one round decreases, exacerbating the degree of data heterogeneity. In Table 2, FedGMT experiences a smaller drop in accuracy when the participation ratio changes from 20% to 5% compared to others.

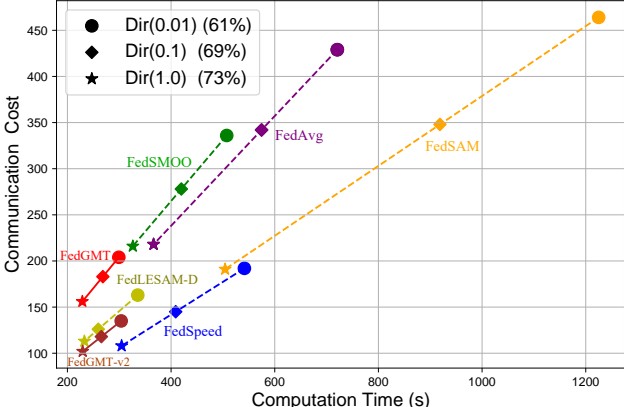

*Figure 4.* Communication cost vs. computation time of FedGMT, FedAvg and other SAM-based methods that achieve a target accuracy in brackets (%). Every connected line represents a method that trains under different Dirichlet settings on CIFAR-10.

**Communication and Computation.** Table 2 presents the communication rounds and computation time needed to achieve a target accuracy. Clearly, FedGMT achieves fewer communication rounds and requires less computation time. SOTA SAM-based methods, FedSpeed and FedSMOO,

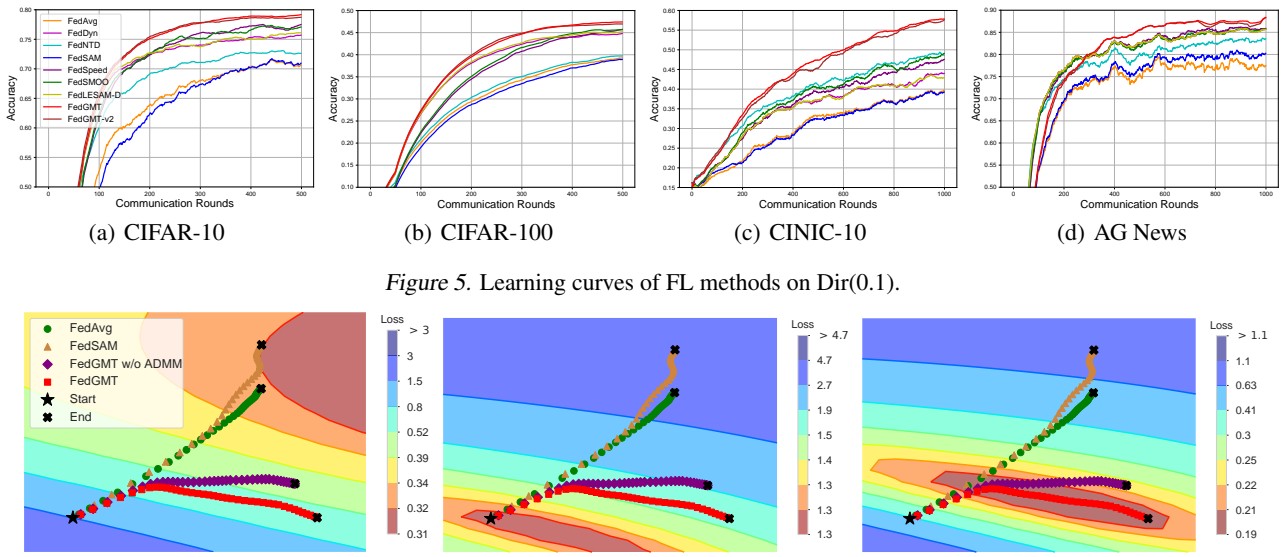

*Figure 5.* Learning curves of FL methods on Dir(0.1).

(a) Local Loss View  (b) Global Loss View  (c) Local & Global Loss View

*Figure 6.* 2D visualization of local learning trajectory on CIFAR-10 (Dir(0.01)). (a): The loss contour is based on one client's local dataset. (b): The loss contour is based on the global dataset. (c): The loss contour is based on the merge of normalized (a) and (b).

both require more than $3.5\times$ in CINIC-10 and $2\times$ in AG News computation time compared to FedGMT. Moreover, we compare the communication cost (i.e., communication round * cost in Table 1) for SAM-based methods in Figure 4. Although FedGMT and FedSMOO achieve better accuracy, they incur higher communication costs. We show that FedGMT-v2 can achieve fewer communication cost, while being less affected by the degree of data heterogeneity, making it more suitable for real-world practicality.

**Learning Curves.** To offer deeper insights into the learning process, we depict the learning curves of various FL methods in Figure 5. For clear visualization, we smooth these curves. Despite employing different communication rounds for each dataset, the model's performance reaches saturation at the end of communication rounds. Across all datasets, both FedGMT and FedGMT-v2 not only attains a superior final model by the end of the communication round but also exhibits much faster convergence compared to the other baselines.

### 4.3. Further Analysis

**Impact of Each Component.** In Table 3, We conduct an ablation study to evaluate the contribution of each component in FedGMT. The results show that both ADMM and $\mathcal{L}^{glotra}$ contribute to performance improvement. It is worth noting that when under high data heterogeneity (e.g. Dir(0.01)), utilizing only $\mathcal{L}^{glotra}$ is ineffective, but when combined with ADMM, the performance significantly improves by 13.39%. The results validate the correctness and effectiveness of our algorithm design and theoretical analysis.

*Table 3.* Ablation study on CIFAR-10.

| ADMM | $\mathcal{L}_m^{glotra}$ | Dir(1) | Dir(0.1) | Dir(0.01) |
|---|---|---|---|---|
| - | - | $75.73_{(0.85)}$ | $70.61_{(3.51)}$ | $61.94_{(4.93)}$ |
| - | $\checkmark$ | $78.18_{(0.43)}$ | $72.68_{(2.19)}$ | $61.28_{(3.11)}$ |
| $\checkmark$ | - | $78.13_{(0.34)}$ | $75.97_{(1.09)}$ | $68.99_{(3.23)}$ |
| $\checkmark$ | $\checkmark$ | $\mathbf{80.48}_{(0.18)}$ | $\mathbf{79.17}_{(0.49)}$ | $\mathbf{74.67}_{(0.77)}$ |

**Local Update Direction.** In this part, we provide an in-depth analysis of "*How does FedGMT guide local learning and why effective?*" To achieve this, we visualize the learning trajectory of the local model on one client utilizing the visualization method (Garipov et al., 2018). Specifically, we train the global model with FedGMT for 100 rounds. Subsequently, we choose one client to perform the loss functions of FedAvg, FedSAM, FedGMT without the ADMM, and FedGMT for 5 local epochs, respectively. Their trajectories are then plotted on the 2D loss landscape with different loss views in Figure 6.

We observe that both FedAvg and FedSAM update towards the local minimum, and FedSAM can explore a flatter region than FedAvg from the local loss view. However, in Figure 6 (b), their directions result in a significant increase in global loss, explaining their unstable performance under high data heterogeneity. In contrast, FedGMT steers each local update towards minimizing the change in global loss. From Figure 6 (c), FedGMT can find a region of joint minimum for both local and global objectives, ensuring that local updates align with the global update. Moreover, the ADMM in FedGMT corrects the update direction towards reducing the joint loss, which can guarantee the effectiveness of the global sharpness measure.

*Table 4.* The test accuracy (%) of `FedGMT` with different hyperparameters on CIFAR10.

| Items | $\alpha$=0.95 | | | | | | $\gamma$=1.0 | | | | | |
|---|---|---|---|---|---|---|---|---|---|---|---|---|
| | $\gamma = 0$ | $\gamma = 0.5$ | $\gamma = 1.0$ | $\gamma = 1.5$ | $\gamma = 2.0$ | $\gamma = 2.5$ | $\alpha = 0$ | $\alpha = 0.5$ | $\alpha = 0.9$ | $\alpha = 0.95$ | $\alpha = 0.995$ | $\alpha = 0.9995$ |
| Dir(1.0) | 78.13 | 79.46 | 80.48 | 80.37 | 80.82 | **80.93** | 78.73 | 79.10 | 79.42 | 80.48 | 81.12 | **81.25** |
| Dir(0.1) | 75.97 | 78.97 | 79.17 | **79.45** | 79.24 | 79.30 | 77.24 | 78.28 | 78.18 | **79.17** | 77.68 | 77.06 |
| Path(4) | 80.38 | 82.07 | **82.79** | 82.38 | 82.67 | 82.57 | 81.42 | 81.76 | 82.47 | **82.79** | 81.76 | 80.09 |
| Path(2) | 75.21 | 78.11 | 78.04 | 77.79 | **78.13** | 77.94 | 78.21 | 78.23 | **78.42** | 78.04 | 75.71 | 73.06 |

(a) FedSpeed (acc = 79.28%)    (b) FedSMOO (acc = 79.10%)    (c) FedLESAM-D(acc = 77.50%)    (d) FedGMT (acc = 79.90%)

*Figure 7.* Visualization of the loss landscapes of the global model trained on CIFAR10-Dir(0.1).

**Loss Landscape.** In Figure 7, we visualize the loss landscapes of global models obtained from `FedSMOO`, `FedSpeed`, `FedLESAM-D` and `FedGMT` after 500 communication rounds. The smoother of the loss landscape contribute to better generalization and higher accuracy, as seen with `FedGMT`, while less optimal landscapes could result in lower accuracy, as in the case of `FedLESAM-D`. These landscapes demonstrate `FedGMT`'s effectiveness in smoothing the loss landscape among SAM-based methods.

### 4.4. Hyperparameters Sensitivity

**Sharpness strength** $\gamma$**.** In Table 4, we compare the performance of the proposed `FedGMT` with different hyperparameters on the CIFAR-10 dataset with different heterogeneous settings. From the results, `FedGMT` exhibits insensitivity to the sharpness strength $\gamma$. Notably, without $\mathcal{L}^{glotra}$ (i.e., $\gamma = 0$) in local learning, there is a significant accuracy drop of 3.2% and 2.83% compared to $\gamma = 1.0$ on Dir(0.1) and Path(2), respectively. Since the improvement from $\gamma = 1.0$ to $\gamma = 2.5$ is negligible, we set $\gamma = 1$ for `FedGMT` and it works well across all datasets. We recommend that the selection of $\gamma$ from $\{0.5, 1.0, 2.0\}$.

**EMA coefficient** $\alpha$**.** The $\alpha$ cannot be selected too large, as it results in significant information loss on the recent update trajectory, particularly under high data heterogeneity (e.g., Dir(0.1), Path(2)). We set $\alpha = 0.95$ for `FedGMT` and it also performs well across all datasets. We recommend that the selection of $\alpha$ from $\{0.95, 0.995, 0.998\}$. Since the global model's performance in FL is unstable during initial epochs, deploying a larger $\alpha$ may impede convergence. In our future work, we plan to implement an adaptive mechanism to further improve performances and reduce sensitivity of $\alpha$.

### 5. Conclusion

In this work, we take a closer look at sharpness-aware minimization in heterogeneous FL from a global view. We propose a novel method `FedGMT` to directly reduce the sharpness of the global model in the FL framework. Theoretical analysis guarantees that `FedGMT` achieves the fast convergence rate of $\mathcal{O}(1/T)$. The extensive experiments demonstrate `FedGMT` achieves high generalization with fewer communication rounds and less computation cost. This work inspires the design of FL frameworks to prioritize reducing the sharpness of the global model.

### Acknowledgments

This work is partially supported by Natural Science Foundation of Shanghai under Grant 24ZR1421500.

### Impact Statement

Our work focuses on the optimization and generalization of federated learning and proposes a novel FL algorithm without compromising data privacy. The insight behind this work may inspire new researches. Since FL has wide applications in machine learning, Internet of Things, and Edge AI, our work may be useful in these areas.

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

# Appendix

**We provide details omitted in the main paper.**

- **Appendix A:** Table of Notations throughout the paper.

- **Appendix B:** Related work.

- **Appendix C:** Derivation of Equation (7) and (8).

- **Appendix D:** Proof of Theorem 3.5 (cf. Section 3.4 of the main paper).

- **Appendix E:** Details of experimental setups.

# A. Table of Notations

*Table 5.* Table of Notations throughout the paper.

| | |
|---|---|
| **FL:** | |
| $M, N$, and $m$ | total number, sampled number, and index of clients |
| $\mathbb{M}, \mathbb{N}^t$ | whole client set, sampled client set at round $t$ |
| $\mathcal{D}, \mathcal{D}_m$ | whole dataset, local dataset |
| $T, t$ | total number and index of communication rounds |
| $w^t, w_m^t$ | global model parameters, local model parameters at round $t$ |
| $C, c$ | total number and index of classes |
| $\eta_g, \eta$ | global learning rate, local learning rate |
| $f$ | neural network |
| $K$ | local interval |
| **SAM**: | |
| $\epsilon, \epsilon_m$ | weight perturbation allocated to global function $\mathcal{L}$, local funcion $\mathcal{L}_m$ |
| $\rho$ | ascent step learning rate in SAM's objective |
| $\nabla$ | abbreviation for $\nabla_w$ on parameters $w$ |
| $S(w)$ | sharpness measure of SAM (equal to $\mathcal{L}(w + \epsilon^*) - \mathcal{L}(w)$) |
| $\Delta^t$ | approximated global gradient at round $t$ ( $\Delta^t = w^t - w^{t+1}$) |
| **Algorithm:** | |
| $\alpha$ | decay coefficient of EMA |
| $\beta$ | penalty coefficient |
| $\gamma$ | strength of minimizing sharpness |
| $\tau$ | temperature in KL function |
| $e^t$ | EMA model |
| $u_m^t, u^t$ | dual variable on local, global |
| **Functions:** | |
| $\mathcal{L}_m$ | local ERM for client $m$ |
| $\mathcal{L}_m^{glotra}$ | global trajectory loss for client $m$ |
| $\mathscr{L}_m$ | loss function of `FedGMT` for client $m$ |
| $\mathscr{F}_m$ | function for convergence analysis |

## B. Related Work

The widely known classical FL method `FedAvg` (McMahan et al., 2017) learns a single global model for all clients by aggregating their local models. Although `FedAvg` provides a practical solution, it still suffers from the heterogeneity of data across clients (Li et al., 2022). With further studies in FL, this is summarized as client drifts (Karimireddy et al., 2020) and inconsistency objectives among clients (Wang et al., 2020b; Shi et al., 2023).

To alleviate the limitations of `FedAvg`, many methods utilize the parameter (or gradient) difference between the local and global model to assist the local training. By incorporating the global model information into local training, the bias between the local and global objectives can be diminished at some level. These approaches can be divided into the following three directions: ① Many algorithmic solutions in (Li et al., 2020; Karimireddy et al., 2020; Acar et al., 2021; Gong et al., 2022; Wang et al., 2023) mainly focus on mitigating the inconsistency across clients via giving a variety of proximal terms to control the local model updates close to the global model. ② Momentum based methods (Xu et al., 2021; Qu et al., 2022; An et al., 2023) introduce the global momentum information into the local training directly, which can force local updates to be similar. ③ Techniques based on knowledge distillation (Lee et al., 2022; Yao et al., 2023; Chen et al., 2023) utilize the global model to guide local models, preventing them from forgetting knowledge from the server. While these algorithms accelerate convergence, they often encounter challenges in sharp global landscapes under high data heterogeneity, leading to unreliable minima and poor stability (Qu et al., 2022; Sun et al., 2023a). Therefore, the global model may not be efficient for all clients, leading to a significant deviation.

Recently, a new research direction in FL has explored the generalization of the global model by analyzing the loss landscape, establishing connections with convergence to flat minima (Mendieta et al., 2022; Qu et al., 2022; Caldarola et al., 2022; Hu et al., 2024a;b;c). Most studies in this direction aim to seek flat minima of the global model with higher generality, utilizing the recently proposed Sharpness-Aware Minimization (SAM) (Foret et al., 2020) optimizer as the local optimizer. Existing works (Foret et al., 2020; Kwon et al., 2021; Du et al., 2021) demonstrate that the flat loss landscape approached by the SAM optimizer exhibits higher stability and generality. In FL, (Qu et al., 2022) first incorporate the SAM optimizer to inherently enhance consistency. Subsequently, various works such as (Caldarola et al., 2022; Sun et al., 2023a;b; An et al., 2023; Lee et al., 2023; Fan et al., 2024) propose different variants to adopt the SAM optimizer.

## C. Derivation of Equation (7) and (8)

In this section, we refer the analysis in previous research (Ganchev et al., 2007; Du et al., 2022) to derivate the equation.

### C.1. Derivation of Equation (7)

$$
\begin{aligned}
e^t &= \alpha e^{t-1} + (1-\alpha)w^t \\
&= \alpha(\alpha e^{t-2} + (1-\alpha)w^{t-1}) + (1-\alpha)w^t \\
&= \vdots \\
&= \alpha^t e^0 + (1-\alpha)(w^t + \alpha w^{t-1} + \alpha^2 w^{t-2} + \cdots + \alpha^{t-1}w^1).
\end{aligned}
\tag{15}
$$

We recall that:

$$
w^t = w^{t-1} - \Delta^{t-1} = w^{t-2} - \Delta^{t-2} - \Delta^{t-1} = \ldots = w^0 - \sum_{i=0}^{t-1}\Delta^i.
\tag{16}
$$

Substituting Equation (15) into Equation (16), we obtain

$$
\begin{aligned}
e^t &= \alpha^t e^0 + (1-\alpha)\left(w^0 - \sum_{i=0}^{t-1}\Delta^i + \alpha\left(w^0 - \sum_{i=0}^{t-2}\Delta^i\right) + \ldots + \alpha^{t-1}(w^0 - \Delta^0)\right) \\
&= \alpha^t e^0 + (1-\alpha)\left(\frac{1-\alpha^t}{1-\alpha}w^0 - \sum_{i=0}^{t-1}\frac{1-\alpha^{t-i}}{1-\alpha}\Delta^i\right) \\
&= \alpha^t e^0 + (1-\alpha^t)w^0 - \sum_{i=0}^{t-1}(1-\alpha^{t-i})\Delta^i \\
&= w^0 - \sum_{i=0}^{t-1}(1-\alpha^{t-i})\Delta^i \\
&= w^t + \sum_{i=0}^{t-1}\alpha^{t-i}\Delta^i.
\end{aligned}
\tag{17}
$$

## C.2. Derivation of Equation (8)

$$
\arg\min_{w^t} S_{\mathbb{N}^t}(w^t) = \arg\min_{\theta^t} S_{\mathbb{N}^t}(\theta^t) \overset{(a)}{=} \arg\min_{\theta^t} cos(\Gamma^t)\|\Delta_{\mathbb{N}^t}(\theta^t)\|\|\Delta_{\mathbb{N}^t}(\theta^t)\|
$$

$$
\overset{(b)}{=} \arg\min_{\theta^t} \left[ cos(\Gamma^t)\|\Delta_{\mathbb{N}^t}(\theta^t)\|\|\Delta_{\mathbb{N}^t}(\theta^t)\| + \sum_{i<t} \alpha^{t-i}cos(\Gamma^i)\|\Delta_{\mathbb{N}^t}(\theta^i)\|\|\Delta_{\mathbb{N}^i}(\theta^i)\| \right]
$$

$$
= \arg\min_{\theta^t} \sum_{i=0}^{t} \alpha^{t-i}cos(\Gamma^i)\|\Delta_{\mathbb{N}^t}(\theta^i)\|\|\Delta_{\mathbb{N}^i}(\theta^i)\|
$$

$$
= \arg\min_{\theta^t} \mathbb{E}_{\theta^i \sim U(\Theta)} \left[ \alpha^{t-i}cos(\Gamma^i)\|\Delta_{\mathbb{N}^t}(\theta^i)\|\|\Delta_{\mathbb{N}^i}(\theta^i)\| \right]
$$

$$
= \arg\min_{\theta^t} \mathbb{E}_{\theta^i \sim U(\Theta)} \left[ \alpha^{t-i}\Delta_{\mathbb{N}^t}(\theta^i)^\top \Delta_{\mathbb{N}^i}(\theta^i) \right]
$$

$$
\overset{(c)}{\approx} \arg\min_{\theta^t} \mathbb{E}_{\theta^i \sim U(\Theta)} \left[ \mathcal{L}_{\mathbb{N}^t}(\theta^i) - \mathcal{L}_{\mathbb{N}^t}(\theta^i - \alpha^{t-i}\Delta_{\mathbb{N}^i}(\theta^i)) \right]
$$

$$
= \arg\min_{\theta^t} \mathbb{E}_{\theta^i \sim U(\Theta)} \left[ \mathcal{L}_{\mathbb{N}^t}(\theta^i) - \mathcal{L}_{\mathbb{N}^t}(\theta^{i+1}) \right]
$$

$$
= \arg\min_{\theta^t} \left[ \mathcal{L}_{\mathbb{N}^t}(\theta^0) - \mathcal{L}_{\mathbb{N}^t}(\theta^1) + \cdots + \mathcal{L}_{\mathbb{N}^t}(\theta^{t-1}) - \mathcal{L}_{\mathbb{N}^t}(\theta^t) \right]
$$

$$
= \arg\min_{\theta^t} \left[ \mathcal{L}_{\mathbb{N}^t}(\theta^0) - \mathcal{L}_{\mathbb{N}^t}(\theta^t) \right]
$$

$$
= \arg\min_{w^t} \left[ \mathcal{L}_{\mathbb{N}^t}(e^t) - \mathcal{L}_{\mathbb{N}^t}(w^t) \right], \tag{18}
$$

where $\Gamma^i$ is the angle between the global gradients that are computed based on active client sets $\mathbb{N}^i$ and $\mathbb{N}^t$, $\theta^i \sim U(\Theta)$ means that $\theta^i$ is uniformly distributed in the set $\Theta$. (a) is from Equation (5). (b) is due to the fact that for $i < t$, $\alpha^{t-i}cos(\Gamma^i)\|\Delta_{\mathbb{N}^t}(\theta^i)\|\|\Delta_{\mathbb{N}^i}(\theta^i)\|$ is a constant with respect to the variable $\theta^t$. (c) utilizes a first-order Taylor expansion.

# D. Proof of Theorem 3.5

## D.1. Preliminary

Before proving the theorem, we first introduce some preliminary conclusions used in our proofs.

**Definition D.1.** $\mathcal{L}_m$ is $L$ smooth if

$$\|\nabla \mathcal{L}_m(x) - \nabla \mathcal{L}_m(y)\| \leq L\|x - y\| \ \forall x, y. \tag{19}$$

Smoothness implies the following quadratic bound,

$$\mathcal{L}_m(y) \leq \mathcal{L}_m(x) + \langle \nabla \mathcal{L}_m(x), y - \boldsymbol{x} \rangle + \frac{L}{2}\|y - x\|^2 \quad \forall x, y. \tag{20}$$

**Lemma D.2.** For random variables $x_1, \ldots, x_n$, we have

$$\mathbb{E}\left[\|x_1 + \ldots + x_n\|^2\right] \leq n\mathbb{E}[\|x_1\|^2 + \ldots + \|x_n\|^2]. \tag{21}$$

**Lemma D.3.** For two random variables $x, y$, we have:

$$\|x + y\|^2 \leq \left(1 + \frac{1}{c}\right)\|x\|^2 + (1 + c)\|y\|^2, \tag{22}$$

where $c > 0$ is a constant.

**Lemma D.4.** For two random variables $x, y$, we have:

$$\langle x, y \rangle \leq \frac{1}{2}\|x\|^2 - \frac{1}{2}\|x - y\|^2. \tag{23}$$

## D.2. Proof of Equation (10)

.

$$
\begin{aligned}
\mathcal{L}_m^{glotra}(w_m^t) &= \frac{\gamma}{|\mathcal{D}_m|} \sum_{\xi_i \in \mathcal{D}_m} \ell_{KL}(f(w_m^t; \xi_i), f(e^t; \xi_i)) \\
&\overset{(a)}{\leq} \frac{\gamma}{|\mathcal{D}_m|} \sum_{\xi_i \in \mathcal{D}_m} \frac{\|f(e^t; \xi_i) - f(w_m^t, \xi_i)\|^2}{\min_{j \in \{1, \cdots, C\}} f(w_m^t, \xi_i)} \\
&\overset{(b)}{\leq} \frac{\gamma L_f}{\delta} \left\|e^t - w_m^t\right\|^2 \\
&\overset{(c)}{\leq} \frac{\gamma L_f}{\delta} \left\|w^t + \sum_{i=0}^{t-1} \alpha^{t-i}\Delta^i - w_m^t\right\|^2 \overset{(d)}{=} \frac{1}{2\beta} \left\|w^t - w_m^t + \sum_{i=0}^{t-1} \alpha^{t-i}\Delta^i\right\|^2,
\end{aligned}
\tag{24}
$$

where (a) use the technique that used in (Yao et al. (2023), Lemma 3), (b) is based on Assumption 3.2 and 3.4, (c) is based on the relation of $e^t$ and $w^t$ in Equation (7). (d): let $\gamma = \frac{\delta}{2\beta L_f}$. Thus we complete the proof.

## D.3. Proof of Theorem 3.5

For convenience, we rewrite our $\mathtt{FedGMT}$ local objective as

$$\mathscr{L}_m^{'}(w_m^t) = \mathcal{L}_m(w_m^t) + \frac{1}{2\beta} \left\| w^t + \sum_{i=0}^{t-1} \alpha^{t-i} \Delta^i - w_m^t \right\|^2 - \langle u_m^t, w_m^t \rangle, \tag{25}$$

based on Equation (9) and (10).

We define

$$\mathscr{F}_m(w_m^t) = \mathcal{L}_m(w_m^t) - \langle \frac{1}{\beta} \sum_{i=0}^{t-1} \alpha^{t-i} \Delta^i, w_m^t \rangle. \tag{26}$$

Then the local objective becomes:

$$\mathscr{F}_m(w_m^t) - \langle u_m^t, w_m^t \rangle + \frac{1}{2\beta} \|w_m^t - w^t\|^2. \tag{27}$$

Throughout the proof, we utilize similar techniques as in (Karimireddy et al., 2020; Acar et al., 2021; Sun et al., 2023a). We define a set of variables which are useful in the analysis. With the partial participation training, Algorithm 1 freezes $w_m^t$ if the client $m$ is not active at round $t$. Thus, we define virtual variables $\widetilde{w}$ as:

$$\widetilde{w}_m^t = \arg\min_w \left\{ \mathscr{F}_m(w) - \langle u_m^t, w \rangle + \frac{1}{2\beta} \|w - w^t\|^2 \right\}, \quad m \in \mathbb{M}. \tag{28}$$

The virtual variable $\widetilde{w}$ is based on partial participation. Considering the first-order gradient condition of the objective in Equation (28) as follows:

$$\nabla \mathscr{F}_m(\widetilde{w}_m^t) - u_m^t + \frac{1}{\beta}(\widetilde{w}_m^t - w^t) = 0, m \in \mathbb{M}; \qquad \nabla \mathscr{F}_m(w_m^t) - u_m^t + \frac{1}{\beta}(w_m^t - w^t) = 0, m \in \mathbb{N}^t. \tag{29}$$

We see that $\widetilde{w}_m^t = w_m^t$ if $m \in \mathbb{N}^t$ and $\widetilde{w}_m^t$ does not depend on $\mathbb{N}^t$, which means that $w_m^t$ equals to $\widetilde{w}_m^t$ with probability $\frac{N}{M}$ and maintains $\widetilde{w}_m^{t-1}$ otherwise. Then we consider the update of $u_m^{t+1} = u_m^t - \frac{1}{\beta}(w_m^t - w^t)$, we infer $u_m^{t+1} = \nabla \mathscr{F}_m(w_m^t)$ at round $t$.

In order to distinguish the parameters before and after updating with the penalty term, we define:

$$\overline{w}^{t+1} \triangleq \frac{1}{N} \sum_{m \in \mathbb{N}^t} w_m^t = w^{t+1} + \beta u^{t+1}. \tag{30}$$

Next, we introduce the following Lemma to assist in the proof.

**Lemma D.5.** *Algorithm 1 satisfies*

$$\mathbb{E}\left[\overline{w}^{t+1} - \overline{w}^t\right] = -\frac{\beta}{M} \sum_{m \in \mathbb{M}} \mathbb{E}\left[\nabla \mathscr{F}_m(\widetilde{w}_m^t)\right]. \tag{31}$$

*Proof.*

$$\mathbb{E}\left[\overline{w}^{t+1} - \overline{w}^t\right] \overset{(a)}{=} \mathbb{E}\left[\frac{1}{N} \sum_{m \in \mathbb{N}^t} \left(w_m^t - w^t - \beta\lambda^t\right)\right] \overset{(b)}{=} \mathbb{E}\left[\frac{1}{M} \sum_{m \in \mathbb{M}} \left(\widetilde{w}_m^t - w^t - \beta\lambda^t\right)\right]$$

$$\overset{(c)}{=} \frac{\beta}{M} \sum_{m \in \mathbb{M}} \mathbb{E}\left[u_m^t - u^t - \nabla \mathscr{F}_m(\widetilde{w}_m^t)\right] \overset{(d)}{=} -\frac{\beta}{M} \sum_{m \in \mathbb{M}} \mathbb{E}\left[\nabla \mathscr{F}_m(\widetilde{w}_m^t)\right],$$

where (a) is from definition in Equation (30), (b) is due to each client is selected with probability $\frac{N}{M}$, (c) is from Equation (29) and (d) is due to the relation $u^t = \frac{1}{M} \sum_{m \in \mathbb{M}} u_m^t$. $\qquad\square$

**Lemma D.6.** *Algorithm 1 satisfies*

$$\mathbb{E}\|\overline{w}^{t+1} - \overline{w}^t\|^2 \leq \frac{1}{M} \sum_{m \in \mathbb{M}} \mathbb{E} \left\| \widetilde{w}_m^t - \overline{w}^t \right\|^2. \tag{32}$$

*Proof.*

$$\mathbb{E}\|\overline{w}^{t+1} - \overline{w}^t\|^2 \overset{(a)}{=} \mathbb{E}\|\frac{1}{N} \sum_{m \in \mathbb{N}^t} (w_m^t - \overline{w}^t)\|^2 \overset{(b)}{\leq} \frac{1}{N}\mathbb{E}\left[ \sum_{m \in \mathbb{N}^t} \|w_m^t - \overline{w}^t\|^2 \right] \overset{(c)}{=} \frac{1}{M} \sum_{m \in \mathbb{M}} \mathbb{E} \left\| \widetilde{w}_m^t - \overline{w}^t \right\|^2, \tag{33}$$

where (a) is from Equation (30), (b) is from Lemma D.2 and (c) is due to each client is selected with probability $\frac{N}{M}$. $\qquad\square$

Note that we use $\|\nabla\mathcal{L}(\overline{w}^t)\|$ the global gradient norm as the metric of the convergence analysis of FedGMT. To achieve this, starting from the assumption of smoothness with Equation (20), taking the full expectation on both sides, we have:

$$\begin{aligned}
&\mathbb{E}\left[ \mathcal{L}(\overline{w}^{t+1}) \right] \\
&\leq \mathbb{E}\left[ \mathcal{L}(\overline{w}^t) \right] + \frac{L}{2}\mathbb{E}\|\overline{w}^{t+1} - \overline{w}^t\|^2 + \mathbb{E}\langle \nabla\mathcal{L}(\overline{w}^t), \overline{w}^{t+1} - \overline{w}^t \rangle \\
&\overset{(a)}{=} \mathbb{E}\left[ \mathcal{L}(\overline{w}^t) \right] + \frac{L}{2}\mathbb{E}\|\overline{w}^{t+1} - \overline{w}^t\|^2 + \beta\mathbb{E}\left[ \left\langle \nabla\mathcal{L}(\overline{w}^t), -\frac{1}{m} \sum_{m \in \mathbb{M}} \nabla\mathscr{F}_m(\widetilde{w}_m^t) \right\rangle \right] \\
&\overset{(b)}{\leq} \mathbb{E}\left[ \mathcal{L}(\overline{w}^t) \right] + \frac{L}{2}\underbrace{\mathbb{E}\|\overline{w}^{t+1} - \overline{w}^t\|^2}_{A1} + \frac{\beta}{2}\underbrace{\mathbb{E}\left\| \frac{1}{m} \sum_{m \in \mathbb{M}} \left( \nabla\mathscr{F}_m(\widetilde{w}_m^t) - \nabla\mathcal{L}(\overline{w}^t) \right) \right\|^2}_{A2} - \frac{\beta}{2}\mathbb{E}\|\nabla\mathcal{L}(\overline{w}^t)\|^2, \tag{34}
\end{aligned}$$

where (a) is from Lemma D.5 and (b) is from Lemma D.4.

From Equation (34), we obtain the global gradient norm $\|\nabla\mathcal{L}(\overline{w}^t)\|$ which we want to bound. Next we need to bound A1 and A2 to complete the proof.

For convenience of expression, we define some quantities that we aim to control.

$$\Psi^t = \frac{1}{M} \sum_{m \in \mathbb{M}} \mathbb{E} \left\| \widetilde{w}_m^t - \overline{w}^t \right\|^2, \quad \Phi^t = \frac{1}{M} \sum_{m \in \mathbb{M}} \mathbb{E} \left\| w_m^t - \overline{w}^{t+1} \right\|^2, \quad Z^t = \sum_{i=0}^{t} \alpha^{t+1-i} \Delta^i.$$

Here, $\Psi^t$ keeps track of how much local models change compared to the average of client models from the previous round, representing the average local updates, $\Phi$ tracks how well local models approximate the current active client average, representing the inconsistency among clients, $Z^t$ tracks the historical global update, representing the cumulative global update with $\alpha$ decay. Note that if models converge, $\Psi^t$ and $\Phi$ will be 0 and $Z^t$ approaches 0 since $\alpha \in (0, 1)$.

**Lemma D.7.** *Based on the assumptions, the A1 and $\Psi^t$ term could be bounded as:*

$$\mathbb{E}\|\overline{w}^{t+1} - \overline{w}^t\|^2 \overset{(a)}{\leq} \Psi^t \leq 10\beta^2 L^2 \Phi^{t-1} + 5\beta^2 L^2 \Psi^t + 5\beta^2\mathbb{E}\|\nabla\mathcal{L}(\overline{w}^t)\|^2 + 5\mathbb{E}\|Z^{t-1}\|^2. \tag{35}$$

*Proof.*

$$
\begin{aligned}
\mathbb{E}\|\overline{w}^{t+1} - \overline{w}^t\|^2 &\overset{(a)}{\leq} \Psi^t = \frac{1}{M} \sum_{m \in \mathbb{M}} \mathbb{E}\left\|\widetilde{w}_m^t - \overline{w}^t\right\|^2 \overset{(b)}{=} \frac{1}{M} \sum_{m \in \mathbb{M}} \mathbb{E}\|\widetilde{w}_m^t - w^t - \beta u^t\|^2 \\
&\overset{(c)}{=} \frac{\beta^2}{M} \sum_{m \in \mathbb{M}} \mathbb{E}\|u_m^t - \nabla\mathscr{F}_m(\widetilde{w}_m^t) - u^t\|^2 \overset{(d)}{=} \frac{\beta^2}{M} \sum_{m \in \mathbb{M}} \mathbb{E}\|\nabla\mathscr{F}_m(w_m^{t-1}) - \nabla\mathscr{F}_m(\widetilde{w}_m^t) - u^t\|^2 \\
&\overset{(e)}{=} \frac{\beta^2}{M} \sum_{m \in \mathbb{M}} \mathbb{E}\|\nabla\mathcal{L}_m(w_m^{t-1}) - \frac{1}{\beta}\sum_{i=0}^{t-2}\alpha^{t-1-i}\Delta^i - \nabla\mathscr{F}_m(\widetilde{w}_m^t) - \nabla\mathcal{L}(w_m^{t-1}) + \frac{1}{\beta}\sum_{i=0}^{t-2}\alpha^{t-1-i}\Delta^i\|^2 \\
&= \frac{\beta^2}{M} \sum_{m \in \mathbb{M}} \mathbb{E}\|\nabla\mathcal{L}_m(w_m^{t-1}) - \nabla\mathcal{L}(w_m^{t-1}) - \nabla\mathscr{F}_i(\widetilde{w}_m^t) + \nabla\mathcal{L}(\overline{w}^t) - \nabla\mathcal{L}(\overline{w}^t)\|^2 \\
&= \frac{\beta^2}{M} \sum_{m \in \mathbb{M}} \mathbb{E}\|\nabla\mathcal{L}_m(w_m^{t-1}) - \nabla\mathcal{L}_m(\overline{w}^t) + \nabla\mathcal{L}_m(\overline{w}^t) - \nabla\mathcal{L}(w_m^{t-1}) + \nabla\mathcal{L}(\overline{w}^t) - \nabla\mathcal{L}_m(\widetilde{w}_m^t) \\
&\quad + \frac{1}{\beta}\sum_{i=0}^{t-1}\alpha^{t-i}\Delta^i - \nabla\mathcal{L}(\overline{w}^t)\|^2 \\
&\overset{(f)}{\leq} \frac{5\beta^2}{M} \sum_{m \in \mathbb{M}} \mathbb{E}\|\nabla\mathcal{L}_m(w_m^{t-1}) - \nabla\mathcal{L}_m(\overline{w}^t)\|^2 + \frac{5\beta^2}{M} \sum_{m \in \mathbb{M}} \mathbb{E}\|\nabla\mathcal{L}_m(\overline{w}^t) - \nabla\mathcal{L}_m(\widetilde{w}_m^t)\|^2 \\
&\quad + \frac{5\beta^2}{M} \sum_{m \in \mathbb{M}} \mathbb{E}\|\nabla\mathcal{L}_m(\overline{w}^t) - \nabla\mathcal{L}_m(w_m^{t-1})\|^2 + 5\beta^2\mathbb{E}\|\nabla\mathcal{L}(\overline{w}^t)\|^2 + 5\mathbb{E}\|\sum_{i=0}^{t-1}\alpha^{t-i}\Delta^i\|^2 \\
&\overset{(g)}{\leq} \frac{10\beta^2 L^2}{M} \sum_{m \in \mathbb{M}} \mathbb{E}\|w_m^{t-1} - \overline{w}^t\|^2 + \frac{5\beta^2 L^2}{M} \sum_{m \in \mathbb{M}} \mathbb{E}\|\overline{w}^t - \widetilde{w}_m^t\|^2 + 5\beta^2\mathbb{E}\|\nabla\mathcal{L}(\overline{w}^t)\|^2 + 5\mathbb{E}\|\sum_{j=0}^{t-1}\alpha^{t-j}\Delta^j\|^2 \\
&= 10\beta^2 L^2 \Phi^{t-1} + 5\beta^2 L^2 \Psi^t + 5\beta^2\mathbb{E}\|\nabla\mathcal{L}(\overline{w}^t)\|^2 + 5\mathbb{E}\|Z^{t-1}\|^2,
\end{aligned}
$$

where (a) is from Lemma D.6, (b) is from Equation (30), (c) is from Equation (29), (d) and (e) is due to the relation $u^t = \frac{1}{M}\sum_{m \in \mathbb{M}} u_m^t$, $u_m^{t+1} = \nabla\mathscr{F}_m(w_m^t)$, the definition of $\mathscr{F}_m$ in Equation (26) and the definition of $\mathcal{L}$ in Equation 1, (f) is from Lemma D.2 and (g) is from L-smooth. $\qquad\square$

**Lemma D.8.** *Based on the assumptions, the A2 term could be bounded as:*

$$
\mathbb{E}\left\|\frac{1}{M} \sum_{m \in \mathbb{M}} \left(\nabla\mathscr{F}_m(\widetilde{w}_m^t) - \nabla\mathcal{L}(\overline{w}^t)\right)\right\|^2 \leq 2L^2\Psi^t + \frac{2}{\beta^2}\mathbb{E}\|Z^{t-1}\|^2. \tag{36}
$$

*Proof.*

$$
\begin{aligned}
\mathbb{E}\left\|\frac{1}{M} \sum_{m \in \mathbb{M}} \left(\nabla\mathscr{F}_m(\widetilde{w}_m^t) - \nabla\mathcal{L}(\overline{w}^t)\right)\right\|^2 &\overset{(a)}{\leq} \frac{1}{M} \sum_{m \in \mathbb{M}} \mathbb{E}\left\|\nabla\mathscr{F}_m(\widetilde{w}_m^t) - \nabla\mathcal{L}(\overline{w}^t)\right\|^2 \\
&\overset{(b)}{\leq} \frac{1}{M} \sum_{m \in \mathbb{M}} \mathbb{E}\left\|\nabla\mathcal{L}_m(\widetilde{w}_m^t) - \nabla\mathcal{L}(\overline{w}^t) - \frac{1}{\beta}\sum_{i=0}^{t-1}\alpha^{t-i}\Delta^i\right\|^2 \\
&\overset{(c)}{\leq} \frac{2}{M} \sum_{m \in \mathbb{M}} \mathbb{E}\left\|\nabla\mathcal{L}_m(\widetilde{w}_m^t) - \nabla\mathcal{L}(\overline{w}^t)\right\|^2 + \frac{2}{\beta^2}\mathbb{E}\left\|\sum_{i=0}^{t-1}\alpha^{t-i}\Delta^i\right\|^2 \\
&\overset{(d)}{\leq} \frac{2L^2}{M} \sum_{m \in \mathbb{M}} \mathbb{E}\left\|\widetilde{w}_m^t - \overline{w}^t\right\|^2 + \frac{2}{\beta^2}\mathbb{E}\|\sum_{i=0}^{t-1}\alpha^{t-i}\Delta^i\|^2 \\
&= 2L^2\Psi^t + \frac{2}{\beta^2}\mathbb{E}\|Z^{t-1}\|^2,
\end{aligned}
$$

where (a) and (c) are from Lemma D.2, (b) is from the definition of $\mathscr{F}_m$ in Equation (26) and (d) is from L-smooth. $\qquad\square$

We see that the result of Equation (35) and (36) have the terms $\Phi^{t-1}$ and $\mathbb{E}\|Z^{t-1}\|^2$. Next, we aim to bound this two terms.

**Lemma D.9.** *Based on the assumptions, the $\mathbb{E}\|Z^t\|^2$ term could be bounded as:*

$$\mathbb{E}\|Z^t\|^2 \le 3\alpha^2\mathbb{E}\|Z^{t-1}\|^2 + 3\alpha^2\mathbb{E}\|Z^{t-2}\|^2 + 27\alpha^2\beta^2G^2 \tag{37}$$

*Proof.* Let $Z^t = \sum_{i=0}^{t}\alpha^{t+1-i}\Delta^i$, then we have

$$Z^t = \alpha Z^{t-1} + \alpha\Delta^t = \alpha Z^{t-1} + \alpha(w^t - w^{t+1}) \tag{38}$$

Therefore, we have

$$
\begin{aligned}
\mathbb{E}\|Z^t\|^2 &= \mathbb{E}\|\alpha Z^{t-1} + \alpha(w^t - w^{t+1})\|^2 \\
&\overset{(a)}{=} \mathbb{E}\|\alpha Z^{t-1} + \alpha(\overline{w}^t - \beta u^t - \overline{w}^{t+1} + \beta u^{t+1})\|^2 \\
&\overset{(b)}{=} \mathbb{E}\|\alpha Z^{t-1} + \alpha(\overline{w}^t - \beta\nabla\mathcal{L}(w_m^{t-1}) + Z^{t-2} - \overline{w}^{t+1} + \beta\nabla\mathcal{L}(w_m^t) - Z^{t-1})\|^2 \\
&\overset{(c)}{\le} \alpha^2\mathbb{E}\|\overline{w}^t - \overline{w}^{t+1} + Z^{t-2} + 2\beta G\|^2 \\
&\overset{(d)}{=} \alpha^2\mathbb{E}\|\frac{\beta}{m}\sum_{m\in\mathbb{M}}\nabla\mathscr{F}_m(\widetilde{w}_m^t) + Z^{t-2} + 2\beta G\|^2 \\
&\overset{(e)}{=} \alpha^2\mathbb{E}\|\beta\nabla\mathcal{L}_m(\widetilde{w}_m^t) - Z^{t-1} + Z^{t-2} + 2\beta G\|^2 \\
&\overset{(f)}{\le} \alpha^2\mathbb{E}\|-Z^{t-1} + Z^{t-2} + 3\beta G\|^2 \\
&\overset{(g)}{\le} 3\alpha^2\mathbb{E}\|Z^{t-1}\|^2 + 3\alpha^2\mathbb{E}\|Z^{t-2}\|^2 + 27\alpha^2\beta^2G^2,
\end{aligned}
$$

where (a) is from Equation (30), (b) and (e) due to the relation $u^t = \frac{1}{M}\sum_{m\in\mathbb{M}}u_m^t$, $u_m^{t+1} = \nabla\mathscr{F}_m(w_m^t)$, the definition of $\mathscr{F}_m$ in Equation (26) and the definition of $\mathcal{L}$ in Equation 1, (c) and (f) is from Assumption 3.3, (d) employs the same proof procedure used in Lemma D.5 and (g) is from Lemma D.2. $\qquad\square$

Following Equation (37), we can obtain that $\mathbb{E}\|Z^0\|^2 \le 4\alpha^2\beta^2G^2$ and $\mathbb{E}\|Z^1\|^2 \le 2\alpha^2\beta^2G^2(9 + 4\alpha^2)$. Then we can get for $t \ge 2$:

$$
\begin{aligned}
\mathbb{E}\|Z^t\|^2 &\overset{(a)}{\le} 27\alpha^2\beta^2G^2\left(\sum_{t=2}^{t}(\sqrt{6}\alpha)^{t-2} + 1\right) + \zeta \\
&\overset{(b)}{\le} 27\alpha^2\beta^2G^2\left(\frac{1}{1 - \sqrt{6}\alpha} + 1\right) + \zeta \\
&= \sigma_G + \zeta \approx \sigma_G,
\end{aligned}
\tag{39}
$$

where (a) utilizes the result of Equation (37) with $\mathbb{E}\|Z^0\|^2$ and $\mathbb{E}\|Z^1\|^2$ to obtain the solution of this inequality. Here, $\zeta$ is a small constant with exponential decay (omitted for brevity). (b) applies the sum of geometric series with $\alpha < \frac{1}{\sqrt{6}}$. We use $\sigma_G$ to denote the result of Equation (39) for convenience.

**Lemma D.10.** *Based on the assumptions, the $\Phi^t$ term could be bounded as:*

$$\Phi^t \le \left(\frac{2N}{2M-N} + \frac{2M}{N}\right)\Psi^t + \frac{2M-2N}{2M-N}\Phi^{t-1} \tag{40}$$

*Proof.*

$$
\begin{aligned}
\Phi^t &= \frac{1}{M}\sum_{m\in\mathbb{M}}\mathbb{E}\|w_m^t - \overline{w}^{t+1}\|^2 = \frac{1}{M}\sum_{m\in\mathbb{M}}\mathbb{E}\|w_m^t - \overline{w}^t + \overline{w}^t - \overline{w}^{t+1}\|^2 \\
&\overset{(a)}{\leq} \left(1 + \frac{N}{2M-N}\right)\frac{1}{m}\sum_{m\in\mathbb{M}}\mathbb{E}\|w_m^t - \overline{w}^t\|^2 + (1 + \frac{2M-N}{N})\mathbb{E}\|\overline{w}^t - \overline{w}^{t+1}\|^2 \\
&\overset{(b)}{=} \frac{N}{M}\left(1 + \frac{N}{2M-N}\right)\frac{1}{M}\sum_{m\in\mathbb{M}}\mathbb{E}\|\widetilde{w}_m^t - \overline{w}^t\|^2 + \left(1 - \frac{N}{M}\right)(1 + \frac{N}{2M-N})\frac{1}{M}\sum_{m\in\mathbb{M}}\mathbb{E}\|w_m^{t-1} - \overline{w}^t\|^2 \\
&\quad + (1 + \frac{2M-N}{N})\mathbb{E}\|\overline{w}^t - \overline{w}^{t+1}\|^2 \\
&\overset{(c)}{\leq} \left[\frac{N}{M}\left(1 + \frac{N}{2M-N}\right) + (1 + \frac{2M-N}{N})\right]\Psi^t + \left(1 - \frac{N}{M}\right)(1 + \frac{N}{2M-N})\Phi^{t-1} \\
&= \left(\frac{2N}{2M-N} + \frac{2M}{N}\right)\Psi^t + \frac{2M-2N}{2M-N}\Phi^{t-1},
\end{aligned}
$$

where (a) is from Lemma D.3, (b) is due to each client is selected with probability $\frac{N}{M}$ and (c) is from Lemma D.6. $\qquad\square$

Collecting and organizing the Equation (34), (35), (36), (39) and (40), we have

$$
\mathbb{E}\left[\mathcal{L}(\overline{w}^{t+1})\right] \leq \mathbb{E}\left[\mathcal{L}(\overline{w}^t)\right] + (\frac{L}{2} + \beta L^2)\Psi^t + \frac{1}{\beta}\sigma_G - \frac{\beta}{2}\mathbb{E}\|\nabla\mathcal{L}(\overline{w}^t)\|^2 \tag{41}
$$

$$
(1 - 5\beta^2 L^2)\Psi^t \leq 10\beta^2 L^2 \Phi^{t-1} + 5\beta^2\mathbb{E}\|\nabla\mathcal{L}(\overline{w}^t)\|^2 + 5\sigma_G \tag{42}
$$

$$
\Phi^t \leq \left(\frac{2N}{2M-N} + \frac{2M}{N}\right)\Psi^t + \frac{2M-2N}{2M-N}\Phi^{t-1} \tag{43}
$$

Let Equation (42) multiplied by constant $\mathscr{X}$ and formula (43) multiplied by constant $\mathscr{Y}$, we take the sum of Equation (41), (42) and (43)

$$
\begin{aligned}
\mathbb{E}\left[\mathcal{L}(\overline{w}^{t+1})\right] + \mathscr{X}(1 - 5\beta^2 L^2)\Psi^t + \mathscr{Y}\Phi^t &\leq \mathbb{E}\left[\mathcal{L}(\overline{w}^t)\right] + (\frac{L}{2} + \beta L^2)\Psi^t + \frac{1}{\beta}\sigma_G - \frac{\beta}{2}\mathbb{E}\|\nabla\mathcal{L}(\overline{w}^t)\|^2 \\
&\quad + 10\mathscr{X}\beta^2 L^2\Phi^{t-1} + 5\mathscr{X}\beta^2\mathbb{E}\|\nabla\mathcal{L}(\overline{w}^t)\|^2 + 5\mathscr{X}\sigma_G \\
&\quad + \mathscr{Y}\left(\frac{2N}{2M-N} + \frac{2M}{N}\right)\Psi^t + \mathscr{Y}\frac{2M-2N}{2M-N}\Phi^{t-1} \tag{44}
\end{aligned}
$$

Collecting the like term of $\Psi^t$ and let the constants $\mathscr{X}$ and $\mathscr{Y}$ satisfy:

$$
\mathscr{X}(1 - 5\beta^2 L^2) = \frac{L}{2} + \beta L^2 + \mathscr{Y}\left(\frac{2N}{2M-N} + \frac{2M}{N}\right) \tag{45}
$$

When Equation (45) holds, Equation (44) will be simplified to:

$$
\begin{aligned}
\mathbb{E}\left[\mathcal{L}(\overline{w}^{t+1})\right] + \mathscr{Y}\Phi^t &\leq \mathbb{E}\left[\mathcal{L}(\overline{w}^t)\right] + (\frac{1}{\beta} + 5\mathscr{X})\sigma_G + (5\mathscr{X}\beta^2 - \frac{\beta}{2})\mathbb{E}\|\nabla\mathcal{L}(\overline{w}^t)\|^2 \\
&\quad + (10\mathscr{X}\beta^2 L^2 + \mathscr{Y}\frac{2M-2N}{2M-N})\Phi^{t-1} \tag{46}
\end{aligned}
$$

Furthermore, considering the coefficient of $\Phi$ term, we let the constant $\mathscr{X}$ and $\mathscr{Y}$ satisfy:

$$
\mathscr{Y} = 10\mathscr{X}\beta^2 L^2 + \mathscr{Y}\frac{2M-2N}{2M-N} \tag{47}
$$

According to the Equation (45) and (47), we can get the solution of $\mathscr{X}$ and $\mathscr{Y}$ as:

$$
\mathscr{X} = \frac{L(N^2 + 2\beta L S^2)}{2(N^2 - 40\beta^2 L^2 M^2 - 25\beta^2 L^2 N^2 + 20\beta^2 L^2 MN)} \tag{48}
$$

$$\mathscr{Y} = \frac{5\beta^2 L^2 N(L + 2\beta L^2)(2M - N)}{N^2 - 40\beta^2 L^2 M^2 - 25\beta^2 L^2 N^2 + 20\beta^2 L^2 MN} \tag{49}$$

This proof requires the constants $\mathscr{X}$ and $\mathscr{Y}$ both to be positive. Moreover, the coefficient of the global gradient term must maintain a positive value. Thus the $\beta$ satifies:

$$N^2 - 40\beta^2 L^2 M^2 - 25\beta^2 L^2 N^2 + 20\beta^2 L^2 MN > 0 \tag{50}$$

$$\frac{\beta}{2} - 5\mathscr{X}\beta^2 > 0 \tag{51}$$

We can get the solution of $\beta$ as: $\beta \le \frac{\sqrt{2}N}{\sqrt{135 + 5\sqrt{5}}ML}$.

We can rewrite Equation (46) as:

$$\underbrace{(\frac{\beta}{2} - 5\mathscr{X}\beta^2)}_{\kappa}\mathbb{E}\|\nabla\mathcal{L}(\overline{w}^t)\|^2 \le (\mathbb{E}\left[\mathcal{L}(\overline{w}^t)\right] + \mathscr{Y}\Phi^{t-1}) - (\mathbb{E}\left[\mathcal{L}(\overline{w}^{t+1})\right] + \mathscr{Y}\Phi^t) + (\frac{1}{\beta} + 5\mathscr{X})\sigma_G \tag{52}$$

Let applying $\mathscr{X} \le \frac{1}{10\beta}$ and $\mathscr{Y} \le \frac{20M\beta^2 L^2}{N}$ the above formula can be telescoped as

$$\frac{1}{T}\sum_{t=1}^{T}\mathbb{E}\|\nabla\mathcal{L}(\overline{w}^t)\|^2 \le \frac{1}{\kappa T}\left(\mathcal{L}(\overline{w}^1) - \mathcal{L}^* + \frac{20M\beta^2 L^2}{N}\Phi^0\right) + \frac{1}{\kappa}(\frac{1}{\beta} + 5\mathscr{X})27\alpha^2\beta^2 G^2\left(\frac{1}{1 - \sqrt{6}\alpha} + 1\right) \tag{53}$$

Similar to Qu et al. (2022); Sun et al. (2023a), we select the $\alpha = \mathcal{O}(\frac{1}{\sqrt{T}})$ that the final convergence rate approaches $\mathcal{O}(\frac{1}{T})$, which completes the proofs. For simplicity and convenience, We choose $\alpha \le \frac{1}{\sqrt{6NT}}$, the above formula can be simplified as:

$$\frac{1}{T}\sum_{t=1}^{T}\mathbb{E}\|\nabla\mathcal{L}(\overline{w}^t)\|^2 \le \frac{1}{\kappa T}\left(\mathcal{L}(\overline{w}^1) - \mathcal{L}^* + \frac{20M\beta^2 L^2}{N}\Phi^0 + \frac{18\beta G}{N}\right) = \mathcal{O}\left(\frac{1}{T}\right) \tag{54}$$

This makes Theorem 3.5 hold.

# E. Experimental Setups

## E.1. Environments

The code is implemented by PyTorch-1.13.1 (Paszke et al., 2019) and the overall code structure is based on PFLlib (Zhang et al., 2023c) library with some modifications. All experiments are conducted on a Linux (Ubuntu-20.04.6 LTS) server with one NVIDIA GeForce RTX 4090 GPU.

## E.2. Datasets

To validate our approach, we consider image and text classification task and adopt four widely used datasets, i.e., CIFAR-10/100 (Krizhevsky Alex, 2009), CINIC-10 (Darlow et al., 2018) and AG News (Zhang et al., 2015). Note that CINIC-10 is constructed from ImageNet (Deng et al., 2009) and CIFAR-10, whose samples are very similar but not drawn from identical distributions. Therefore, it naturally introduces distribution shifts which is suited to the heterogeneous nature of federated learning. The details about each datasets and setups are described in Table 6.

*Table 6.* Details datasets setups used in the experiment.

| Datasets | CIFAR-10 | CIFAR-100 | CINIC-10 | AG News |
|---|---|---|---|---|
| Datasets Classes | 10 | 100 | 10 | 4 |
| Datasets Size (train/test) | 50,000/10,000 | 50,000/10,000 | 90,000/90,000 | 120,000/7,600 |
| Number of Clients | 100 | 100 | 100 | 100 |
| Client Sampling Ratio | 0.1 | 0.1 | 0.1 | 0.1 |
| Local Epochs | 5 | 2 | 5 | 5 |
| Batch Size | 50 | 20 | 100 | 50 |
| Learning Rate | 0.01 | 0.01 | 0.01 | 0.1 |
| Base Optimizer | SGD | SGD | SGD | SGD |
| Momentum | 0.9 | 0.9 | 0.9 | 0.9 |
| Weight Decay | 1e-5 | 1e-5 | 1e-5 | 1e-5 |
| Rounds | 500 | 500 | 1000 | 1,000 |

We use standard pre-processing, where CIFAR-10/100 and CINIC-10 images are normalized. The $32 \times 32$ CIFAR-10/100 and CINIC-10 images are padded 2 pixels each side, randomly flipped horizontally, and then randomly cropped back to $32 \times 32$. For AG News, the maximum length of the sentence vector is set to 200.

## E.3. Model Architecture

We conduct further experiments on different model architectures: CNN (LeCun et al., 1998) for CIFAR-10, ResNet-8 (He et al., 2016) for CIFAR-100 and ViT (Dosovitskiy et al., 2021) for CINIC-10 and FastText (Joulin et al., 2017) for AG News.

- The CNN used in our experiment is from `FedAvg` (McMahan et al., 2017), a similar architecture is used in (Luo et al., 2021; Lee et al., 2022).

- The ResNet-8 in our experiment is from PFLlib (Zhang et al., 2023c) library, a similar architecture is used in (Shang et al., 2022a;b). We follow the suggestion of Hsieh et al. (2020) to replace the Batch Normalization (Ioffe & Szegedy, 2015) with the Group Normalization (Wu & He, 2018) to avoid the non-differentiable parameters.

- The ViT is adopted from ViT-CIFAR[1], which is a smaller version compared to the original ViT.

- The FastText is from PFLlib (Zhang et al., 2023c) library, a similar architecture is used in (Zhang et al., 2023b;a).

---

[1]https://github.com/omihub777/ViT-CIFAR

### E.4. Baselines Implementation Details

We consider the following six state-of-the-art FL methods:

- `FedAvg` (McMahan et al., 2017) is proposed as the basic framework in the federated learning, which aggregates the locally trained model parameters by weighted averaging proportional to the amount of local data that each client had.

- `FedDyn` (Acar et al., 2021) dynamically updates its regularizer so that the optimal model for the regularized loss is in conformity with the global empirical loss.

- `FedNTD` (Lee et al., 2022) conducts local-side distillation only for the not-true classes to prevent forgetting.

- `FedSAM` (Qu et al., 2022) directly applies the SAM objective in local learning.

- `FedSpeed` (Sun et al., 2023b) incorporates a local SAM optimizer with a dynamic regularizer.

- `FedSMOO` (Sun et al., 2023a) utilizes the Alternating Direction Method of Multipliers (ADMM) to estimate global perturbation and adopts a dynamic regularizer during the local training.

- `FedLESAM` (Fan et al., 2024) estimates the global perturbation as the difference between the locally stored historical model from the activation round and the global model received in the current round.

For the above algorithms, we search hyperparameters and choose the best among the candidates. All methods hyperparameters is refered from their official implementations or papers. The hyperparameters for each algorithm is in Table 7.

*Table 7.* Algorithm-specific hyperparameters used in the experiment.

| Method | Searched Candidates | Best Selection |
|---|---|---|
| `FedAvg` | None | None |
| `FedDyn` | $\beta \in \{1, 10, 100, 1000\}$ | $\beta = 10$ except $\beta = 100$ in AG News |
| `FedNTD` | None | $\gamma$=1.0, $\tau = 1.0$ |
| `FedSAM` | $\rho \in \{0.5\eta, \eta, 2\eta, 5\eta, 10\eta\}$ | $\rho = 5\eta$ except $\rho = \eta$ in CIFAR-100 |
| `FedSpeed` | $\rho \in \{0.5\eta, \eta, 2\eta, 5\eta, 10\eta\}$ | $\rho = 5\eta$ except $\rho = \eta$ in CIFAR-100 |
|  | $\beta \in \{1, 10, 100, 1000\}$ | $\beta = 10$ except $\beta = 100$ in AG News |
| `FedSMOO` | $\rho \in \{0.5\eta, \eta, 2\eta, 5\eta, 10\eta\}$ | $\rho = 5\eta$ except $\rho = \eta$ in CIFAR-100 |
|  | $\beta \in \{1, 10, 100, 1000\}$ | $\beta = 10$ except $\beta = 100$ in AG News |
| `FedLESAM-D` | $\rho \in \{0.5\eta, \eta, 2\eta, 5\eta, 10\eta\}$ | $\rho = 5\eta$ except $\rho = \eta$ in CIFAR-100 |
|  | $\beta \in \{1, 10, 100, 1000\}$ | $\beta = 10$ except $\beta = 100$ in AG News |
| **FedGMT** | $\gamma \in \{0.5, 1.0, 2.0\}$ | $\gamma = 1.0$, $\tau = 3.0$ |
|  | $\beta \in \{1, 10, 100, 1000\}$ | $\beta = 10$ except $\beta = 100$ in AG News |
|  | $\alpha \in \{0.5, 0.95, 0.995, 0.998\}$ | $\alpha = 0.95$ for FedGMT, $\alpha = 0.5$ for FedGMT-v2 |

In Table 7, $\beta$ is the coefficient for the penalty term. The selection of this hyperparameter has been studied in many previous works which verify its efficiency. Usually the selection of $\beta$ are in $\{10, 100\}$. The selection of $\rho$ is usually related to the local learning rate $\eta$ (Sun et al., 2023b). $\gamma$ and $\tau$ is the strength and temperature in the KL function.

