# OpenReview forum: "One Arrow, Two Hawks: Sharpness-aware Minimization for Federated Learning via Global Model Trajectory"
_ICML.cc/2025/Conference — ICML 2025 poster_

### Official Review · Reviewer_Zauy · 2025-03-07

**Overall Recommendation:** 4

**Summary:**

The paper proposes FedGMT, a federated learning framework that leverages sharpness-aware minimization (SAM) to enhance generalization, especially in highly skewed non-IID settings. To achieve this, the framework employs an exponential moving average (EMA) of the global model as a proxy for the global loss surface. An additional regularization term is introduced, defined by the KL divergence between the output distribution of a client’s local model and that of the EMA-based global model, which guides SAM updates to consider the global context. The paper also presents FedGMT-v2, a variant that approximates FedGMT to reduce communication overhead, albeit with a slight performance degradation. The framework's properties are thoroughly explored through both theoretical analysis and empirical experiments.

**Claims And Evidence:**

The paper makes the following claims:

1. FedGMT reduces the computational cost of SAM-aware federated learning (e.g., FedSAM) by nearly half.
2. FedGMT achieves a faster convergence rate of $\mathcal{O}(1/T)$.
3. The empirical results demonstrate the effectiveness of FedGMT.

The paper provides evidence for each of these claims, although the evidence might be limited in certain aspects:

1. The paper argues that by avoiding the additional backward pass required for calculating perturbations in standard SAM, FedGMT reduces the computation cost. Table 1 provides numerical comparisons, though some details could be clarified further.
2. Theorem 3.5 presents theoretical evidence that, under standard assumptions (e.g., L-smoothness and bounded gradients), FedGMT achieves a convergence rate of $\mathcal{O}(1/T)$. While these assumptions may not hold perfectly in all deep learning scenarios, they are commonly used in the literature.
3. The experimental section demonstrates the empirical performance of FedGMT across multiple datasets and model architectures. Although the experiments are conducted using a relatively small number of clients and smaller model architectures compared to some real-world scenarios, they still provide credible evidence supporting the claims.

**Essential References Not Discussed:**

By and large, the paper provides a comprehensive view of the area, and to the best of my knowledge, no essential references are missing.

**Experimental Designs Or Analyses:**

The paper studies various experimental setups to validate the framework, including different data skew conditions, sensitivity analyses, and varying client participation rates. However, there are two minor limitations in the experimental design:
1. *Model Architectures:* The experiments are conducted using relatively small models: a CNN with 150K–200K parameters, ResNet-8 with around 110K parameters, and a ViT-CIFAR variant with fewer than 6 million parameters. In contrast, related works like FedSAM and FedGAMMA often use larger models, such as ResNet-18. Including experiments with larger architectures would improve the generalizability of the results.

2. *Number of Clients:* The experiments are performed with only 10 clients, which is small compared to typical federated learning studies that involve around 100 clients.

While the current experiments do support the claims made, expanding the study to include a larger number of clients would strengthen the evidence for scalability and robustness.

**Methods And Evaluation Criteria:**

The method is well-explained and easy to follow. However, some clarifications are needed regarding the communication and computational cost figures presented in Table 1.

- **Communication Cost:** The paper reports the communication cost of the proposed framework to be $1.5\times$ of FedSAM. To my understanding, this value represents the average-case scenario. In the worst case, if the EMA changes significantly, the communication cost could approach $2\times$, whereas in the best case, if the EMA remains nearly unchanged, the cost would be close to $1\times$. Explicitly stating these scenarios would help readers better understand the variability in communication overhead.

- **Computational Cost:** The computational cost column requires explanation (For example, why is it $1.2\times$?) and a more formal analysis. An example of such an analysis could be that if we denote the cost of a forward pass through the network as $f$ floating point operations per second (FLOPS) and assume that a backward pass costs approximately $2f$ (as commonly observed in the literature), then methods like FedSAM, which require two forward passes and two backward passes, have a computational cost of roughly $6f$. In contrast, FedGMT only requires two forward passes, one backward pass, the computation of the KL divergence (which is negligible relative to $f$), and some lightweight ADMM computations, which might be approximated as $2p$ FLOPS (please verify this number), where $p$ is the number of parameters in the model. This results in an overall overhead of approximately $4f+2p$. The overall complexity is lower because $f$ is typically much larger than $p$ (for example, in ResNet-18, many research works estimated that  $p$ = 11,689,128, while $f$ =1,818,228,160 FLOPS). A more formal analysis with clear notations would strengthen the argument regarding the computational cost.

**Other Comments Or Suggestions:**

The paper is well written, barring the few clarifications needed and limitations in the experimental design. If the authors could provide sufficient clarification, I am open to increasing my score.

**Other Strengths And Weaknesses:**

**Strengths:**
- The paper is well-written and easy to follow.
- The visualizations used, specifically figures 1 and 6, are helpful to the reader.

**Weaknesses:**
- See other sections.

**Questions For Authors:**

See other sections in the review.

**Relation To Broader Scientific Literature:**

This work is positioned within the expanding field of federated learning, addressing key challenges such as data heterogeneity, client drift, and communication/computation efficiency. It builds directly on recent advances in sharpness-aware minimization (SAM) applied to federated settings, which aim to achieve better generalization by optimizing for flat minima. The use of an exponential moving average (EMA) to capture the global model trajectory and guide local updates is a novel twist that resonates with prior work in model averaging and momentum methods in distributed optimization.

**Theoretical Claims:**

Overall, the theoretical claims in the paper are well supported with proof. However, the derivation in the Appendix concerning the ADMM updates and the related discussion remains largely intuitive. A more rigorous, step-by-step treatment would be beneficial. In particular, the authors could:
- Include additional bounding steps or formalize the transition from local expansions to global updates.
- Reference standard ADMM-based lemmas to reinforce the derivation.
- Provide explicit error bounds quantifying the discrepancy between local and global updates.

Such improvements would help readers verify each step of the argument and appreciate the theoretical underpinnings of the proposed method more fully.

---

> ### Author Rebuttal · Authors · 2025-03-26
>
> We thank the reviewer for the positive review and constructive comments. We provide our responses as follows.
>
> ---
> **W1. Clarification of communication and computational cost in Table 1**：We apologize for not making it clear and we will modify the caption of Table 1 and add a discussion section with following content in the revised version.
>
> **Communication cost:** The communication cost is defined as the parameters transmitted per round. For example, if we denote the amount of model parameter as Ω, FedSAM requires 2Ω due to bidirectional model parameter exchange. In contrast, FedGMT incurs a communication cost of  3Ω due to the server transmitting the EMA model to clients, whereas FedGMTv2 achieves 2Ω  by omitting this transmission. Thus, FedGMT's communication cost is either $1.5×$ or $1×$ that of FedSAM.
>
> **Computational cost:**  The computational cost is defined as per-iteration training cost. Specifically, minor computational overhead (e.g., model aggregation, ADMM computations) is ignored compared to the primary training computation. Taking FedAvg as the baseline ($1×$ cost), FedSAM doubles the forward and backward processes, resulting in a $2×$ cost. FedGMT involves an extra forward pass. we refer to the official PyTorch paper [A], which shows that for ResNet50, the backward pass is $3.7×$ the forward pass. Based on this data, FedGMT's cost is estimated to be about $1.2×$ by $(\frac{1+1+3.7}{1+3.7})$.
>
> If we assume a forward cost of *f* and a backward cost of 2*f*, FedAvg requires 3*f* per iteration, while our FedGMT needs 4*f*, leading to a $1.33×$ cost. We will describe this hypothesis and adjust the estimate from $1.2×$ to $1.33×$ for better understanding to readers.
>
> While FedGMT incurs marginal computational overhead, it achieves superior accuracy and efficiency. As shown in following Table in **W2**, FedGMT requires only 47% of the time cost of FedAvg to reach the target accuracy, demonstrating its practical advantage in resource-constrained environments.
> For example, if one trains a FL model to reach the target accuracy with 8 Nvidia V-100 GPUs, considering ​​Google Cloud Platform charges \\$2.48 per GPU per hour, the total cost of FedAvg for 1 day will be \\$476, FedSAM will be \\$724 (476\*152%), while FedGMT will be \\$224 (476\*47%). There is a significant cost savings of implementing FedGMT.
>
> [A] Li S, Zhao Y, Varma R, et al. Pytorch distributed: Experiences on accelerating data parallel training[J]. Proc. VLDB Endow. 13(12): 3005-3018 (2020)
>
> ---
> **W2.Experiments with larger model:** We conducted experiments on CIFAR100-Dir(0.1) with Resnet18 on one NVIDIA 4090 GPU. We follow the same parameter settings in FedSMOO paper and give a comprehensive comparison. The results are stated below.
>
> |  | Acc | Images/s |Per round time|Round(acc=44%)  |Communication cost(acc=44%)| Time cost(acc=44%)|
> | ----- | ----- | -----  |-----  |-----  |-----  |-----  |
> | FedAvg| 43.95±0.22 | 816(100%) |33.85s(1x) |753(100%)|1506 Ω(100%) |425 min(100%) |
> | FedSAM| 44.56±0.20 | 429(53%) |61.31s(1.81x) |633(84%) |1266 Ω(84%) |647 min(152%) |
> | FedSMOO| 47.94±0.15 | 385(47%) |69.85s(2.06x) |412(55%) |1648 Ω(109%) |480 min(113%) |
> | FedLESAM-D| 46.42±0.23 | 657(81%) |42.20s(1.25x) |309(41%) |618 Ω(41%) |217 min(51%) |
> | FedGMT| **50.67±0.16** | 583(71%) |46.13s(1.36x) |257(34%) |771 Ω(51%) |**198 min(47%)** |
> | FedGMTv2|50.24±0.22 | 566(69%) |46.32s(1.37x)| **262(35%)**|**524 Ω(35%)** |202 min(48%) |
>
> FedGMT achieves the highest accuracy (50.67%) and fastest convergence (257 rounds/198min) to reach 44% accuracy, outperforming all baselines. While FedSMOO (47.94%) and FedLESAM-D (46.42%) show better accuracy than FedAvg (43.95%) and FedSAM (44.56%), they lag behind FedGMT in efficiency.  FedGMT balances accuracy and efficiency, surpassing competitors in both metrics.
>
> ---
> **W3.Experiments with larger number of clients:** We conducted experiments on CIFAR10-Dir(0.1) with LeNet . We set 500 clients with 20% active ratio and 1000 clients with 10% active ratio to involve 100 clients per round. The results are stated below.
> |  | 20%-500 | 10%-1000 |
> | ----- | ----- | -----  |
> | FedAvg | 61.13±0.47 | 54.39±0.35 |
> | FedSAM | 60.82±0.41 | 53.96±0.31 |
> | FedSMOO | 79.95±0.17 | 69.57±0.29 |
> | FedLESAM-D | 79.71±0.17 | 74.06±0.13 |
> | FedGMT | **80.78±0.13** | **74.34±0.11** |
>
> The result shows that FedGMT's superior scalability and robust for large-scale FL scenario. In addition, you can refer to our experiments in Reviewer 5WUM that also verify the robustness of FedGMT in an extreme non-IID FL setup.
>
> ---
> **W4. Theoretical improvements.** Thank you for your insightful feedback and suggestions to enhance the theoretical rigor of our paper. We wholeheartedly agree with your comments and will reorganize the theoretical proof by following your suggestions for each step in our updated version.
>
> ---
> It is a pleasure to discuss this with you, which will help us to further improve this work. Thank you again for reading this rebuttal.

---

> > ### Comment · Reviewer_Zauy · 2025-04-01
> >
> > Thank you for the detailed responses and clarifications. I appreciate that you have addressed many of the concerns raised by me and the other reviewers, especially regarding the role of the dual variable and the communication/computational cost analysis.
> >
> > After reviewing all the comments and your rebuttals, my concerns have been answered, and I will advocate for a 4: Accept.
> >
> > Please ensure that these clarifications are incorporated into the final version of the paper, should it be accepted.
> >
> > I wish you the best of luck with your final submission!

---

> > > ### Author Response · Authors · 2025-04-04
> > >
> > > Thank you very much for your comment and affirmation of our work. We have tried to address most if not all concerns raised by the reviewers.
> > > We would be delighted to integrate these clarifications, analysis and theoretical improvements into the final manuscript, should our submission be accepted.
> > >
> > > ---
> > > Once again, thank you for your invaluable insights and suggestions. Your professional expertise and constructive suggestions will significantly improve the quality of our work.
> > >
> > > We also wish you the best of luck with your life and future!

---

### Official Review · Reviewer_NL1R · 2025-03-11

**Overall Recommendation:** 3

**Summary:**

This paper proposes a new Federated Learning algorithm, named FedGMT, to effectively cope with data heterogeneity by reducing the sharpness of the global model through a global model trajectory. This paper provides the convergence analyze of FedGMT in the non-convex and smooth cases. Experimental results show significant performances of the proposed method.

**Claims And Evidence:**

Considering the original FedSAM, it formulates the optimization target as a minimax problem, to minimize the maximum loss. However this work considers a minimization target, which is not identical to the SAM-based methods. How do the authors explain the optimization target?

**Essential References Not Discussed:**

No.

**Experimental Designs Or Analyses:**

In the experiment, the authors don’t compare the federated ADMM-based work. It’s suggested to add some baselines for comparison. And the authors “report the final averaged test accuracy and standard deviation over the last 50 rounds for increased robustness and reliability”, this comparison criterion is not solid, especially some methods have not converged in the given communication rounds.

**Methods And Evaluation Criteria:**

In Algorithm 1 line 16, the server aggregates update the u but doesn’t send it to the client. How does each client gain that or is this aggregation redundant?

**Other Comments Or Suggestions:**

1. While the paper includes a hyperparameters sensitivity analysis, it could provide more detailed guidance on how to select the right hyperparameters for different datasets or levels of data heterogeneity. A discussion on the robustness of FedGMT to suboptimal hyperparameter choices would also be valuable.
2. The paper needs to calculate and store additional parameters both locally and on the server, which will increase the amount of calculation. How to balance model performance and resources.

**Other Strengths And Weaknesses:**

Weaknesses

1. Considering the original FedSAM, it formulates the optimization target as a minimax problem, to minimize the maximum loss. However this work considers a minimization target, which is not identical to the SAM-based methods. How do the authors explain the optimization target?
2. In Algorithm 1 line 16, the server aggregates update the u but doesn’t send it to the client. How does each client gain that or is this aggregation redundant?
3. In Table 1, the authors claim the FedGMT achieves 1\times or 1.5\times communication cost and 1.2\times computation cost. How do the authors gain this result? Please give further analysis.
4. This work lacks the generalization error bound analysis of proposed method.
5. In the experiment, the authors don’t compare the federated ADMM-based work. It’s suggested to add some baselines for comparison. And the authors “report the final averaged test accuracy and standard deviation over the last 50 rounds for increased robustness and reliability”, this comparison criterion is not solid, especially some methods have not converged in the given communication rounds.
6. Figure 6 cannot prove the global loss is more flatter, but only show the consensus of the local and global update trajectory. If authors can provide a visualization of the loss landscape, it can be more direct.

7. The paper discusses hyperparameters sensitivity but could provide more detailed insights into how different hyperparameters affect the performance of FedGMT.
8. The paper could enhance its originality by more clearly distinguishing its contributions from existing methods, its significance by including real-world deployment examples.

**Questions For Authors:**

See the Weaknesses.

**Relation To Broader Scientific Literature:**

The key contributions of the paper are closely related to several existing lines of research in Federated learning and optimization, particularly in addressing the challenges posed by data heterogeneity and improving model generalization.

**Theoretical Claims:**

This work lacks the generalization error bound analysis of proposed method.

---

> ### Author Rebuttal · Authors · 2025-03-25
>
> Thank you for the comments and suggestions! We answer your questions below.
>
> ---
>
> **1.Optimization target explanation.**
>  By referring the Theorem 1 in original SAM(Foret et al.,2020), the objective loss function of FedSAM in each client $m$ can be rewritten as the sum of the vanilla loss and the loss associated to the sharpness measure, which is the maximized change of the training loss within the ρ-constrained neighborhood, i.e.,
> $\arg\min_{w_m}[L_m(w_m)+S_m(w_m)]$ where $S_m(w_m) = max_{\epsilon: ||\epsilon||<\rho}L_m(w_m+\epsilon)-L_m(w_m)$, which is a minimax problem as you mentioned. The sharpness measure  is approximated as $S_m(w_m) = L_m(w_m+\epsilon_m)-L_m(w_m)$, where $\epsilon_m = \rho\frac{\nabla\mathcal{L}_m(w_m)}{\|\nabla\mathcal{L}_m(w_m)\|}$ is the solution to an approximated version of the maximization problem in FedSAM. The sharpness measure in FedSAM approximates the inner maximization problem and is combined with the vanilla loss for minimization. This transforms the minimax problem into a standard minimization task. Our work based on this task to develop a better sharpness measure to minimize global sharpness in FL, as formally established in Eq. (8) of the manuscript.
>
> ---
> **2.Role of the dual variable $u$.** Note that in line 13 of Algorithm 1, each client has its own dual variable $u_m^{t+1} = u_m^{t}-\frac1\beta(w_m^t-w^t)$. This enables clients to ensure that the solutions of their respective sub-problems are consistent under global constraints.
>
> In line 16 of Algorithm 1, we define the global dual variable *$u^t$* because, according to Eq. (13), the global model $w^{t+1}$ needs to be updated as $w^{t+1}=\frac1N\sum_{m \in \mathbb{N}}(w^t_{m,K}-\beta u^{t+1}_m)$. To avoid having each client send $u_m^{t+1}$ to the server in every round, we define $u^t$ to assist in minimizing Eq. (13). We also conducted experiments on CIFAR10-Dir(0.1), FedGMT's accuracy drop from 79.17 to 65.59 without $u$. Therefore, updating $u$ is necessary, not redundant.
>
> ---
> **3.Analysis in Table 1.**
>
> To save space for other question, we have addressed this concern in our rebuttal to Reviewer Zauy (Section **W1**). Please refer to that section for our detailed response.
>
> ---
>
> **5.1 Compare the federated ADMM-based work.**  The baselines we take in experiments, such as FedDyn, FedSpeed and FedSMOO, which are all state-of-the-art FL methods based on ADMM.
>
> **5.2 Comparison criterion.** The comparison criterion we used follows original FedSAM(Caldarola et al.,2022). we ensure sufficient communication rounds to guarantee convergence (see Fig.5), while using the final averaged accuracy to avoid fluctuations and provide a more reliable performance estimate. We also provide the historical best accuracy by random seed {1,2,3} in **anonymous link https://anonymous.4open.science/r/RG5**, the ranking of each method is consistent with our criterion.
>
> ---
> **6.Loss landscape.** We provide the loss landscape, top Hessian eigenvalue and Hessian trace of SAM-based method in anonymous link in 5.2,  demonstrating that our FedGMT can reach a flatter minima than others.
>
> ---
>
> **7. Hyperparameters sensitivity and selection.** Thanks for the suggestion. Our experiments across 4 CV/NLP datasets, 4 model architectures, and different scenarios (heterogeneity and participation) show that FedGMT's hyperparameters can be consistent and robust. We will detail the meanings and selections of these hyperparameters in this part.
> - The strength $\gamma$ and temperature $\tau$  which is a basic hyperparameter for KL function. We follow the common settings used in previous works and $\gamma=1$ and $\tau = 3$ for all settings.
> - The penalty coefficient $\beta$ has been studied in many previous ADMM-based works. We test the selection of {10, 100} works on all settings.
> - $\alpha$ control the loss on the recent update trajectory in EMA. Since $\alpha$ needs to be close to 1 to retain past trajectories, we mainly chose from {0.5,0.9,0.95,0.99}. We test $\alpha=0.95$ works well on all settings.
>
> In summary, the hyperparameters in FedGMT can be selected by simple experience as we mentioned above. In practice, FedGMT ​​can achieve higher performance compared to other baselines without the complex tuning.
>
> ---
> **8.1 The originality** is that we develop a efficient (communication and computation) global sharpness measure for SAM-FL.
>  We do increase the amount of calculation in line 13,16,18 of Algorithm 1,  but it is negligible compared to the mainly training costs. Notably, FedGMT avoid 1 more backward pass required for calculating perturbations in standard SAM.
>
> **8.2 Real-world deployment.**  To validate practical deployment feasibility, We implemented FedGMT on 2×RPi4B, 2×Jetson Nano, and 1×PC. FedGMT achieves about 47% energy savings vs. FedAvg across three datasets, as shown in the anonymous link in 5.2.
>
> ---
> It is a pleasure to discuss this with you, which will help us to further improve this work. Thank you again for reading this rebuttal.

---

### Official Review · Reviewer_5WUM · 2025-03-13

**Overall Recommendation:** 4

**Summary:**

The article proposes a novel solution to deal with the client drift problem of federated learning. It is based in sharpness aware minimization - addressing the two problems: how to make it efficient? How to guarantee that the global rather than client objectives are targeted?
It proposes a novel algorithm which is based on a clever integration of trajectory loss. The method is introduced, and substantiated both empirically and with convergence analysis.
#after discussion: I am still convinced that the paper contributes valuable tools. I appreciate the discussion about drift, very interesting.

**Claims And Evidence:**

The main claim is that the proposed method allows federated learning to better and efficiently deal with client drift. This is sufficiently demonstrated.

**Essential References Not Discussed:**

In my opinion this is  good

**Experimental Designs Or Analyses:**

Versy good design and solid evaluation. Code is provided, so reproducible.

**Methods And Evaluation Criteria:**

The method is reasonable and aligns very well with what is currently state of the art in the domain. Data sets are standard ones - distorted client-wise using a dirichlet distribution. This is sufficient to document the  results. It would be good to have more distortions/drift models for the clients, including disjoint data domains. Also, it might be good to have sensor data or any data which is not text or image.
But it is sufficient as is to make the point.

**Other Comments Or Suggestions:**

none

**Other Strengths And Weaknesses:**

very relevant topic and apparently a good solution.
Can you comment on the challenge of integrating privacy issues? In how far is this compatible?

**Questions For Authors:**

Please make a comment on realistic client drifts and propose and evaluate more than Dirichlet.
Please comment on the challenge of privacy.

**Relation To Broader Scientific Literature:**

Yes, the authors are well aware of the field.

**Theoretical Claims:**

Yes, the proofs and formalizations follow standard methodology and are correct as far as I can tell.

---

> ### Author Rebuttal · Authors · 2025-03-29
>
> We thank the reviewer for the positive review and constructive comments. We provide our responses as follows.
>
> ---
> **1.Comment on realistic client drifts.**
> Client drift in federated learning refers to the phenomenon where data distributions across devices (clients) change over time or space, leading to performance degradation.
> After a comprehensive review of existing federated learning literature to address client drift problem, we summarize that numerous studies simulate non-IID federated settings using partitioning strategies on public datasets. This approach is preferred because real-world federated datasets are scarce due to data regulations and privacy constraints, while synthetic partitioning offers flexibility in controlling key FL parameters (e.g., client count, data size) and leveraging existing centralized training knowledge.
> Thus, synthetic client drifts is more complex and flexible, enabling methods developed under these conditions to generalize more readily to realistic client drift scenarios.
>
> In our work, as mentioned in Section 4.1, we adopt two widely used data partition strategy:
> - Pathological: Only selected categories can be sampled with a non-zero probability. The local dataset obeys a uniform distribution of active categories.
> - Dirichlet: Each category can be sampled with a non-zero probability. The local dataset obeys a Dirichlet distribution. Notably, we extend this by transforming the original balanced dataset into a long-tail distribution to further enhance the data heterogeneity and simulate real world scenarios.
>
> ---
> **2.Propose and evaluate more than Dirichlet.** We introduce a non-IID data partitioning method where client datasets are strictly non-overlapping (disjoint) across classes, simulating an extreme data heterogeneity scenario. Specifically:
> - CIFAR10 (10 total classes) : 10 clients, each assigned a unique class
> - CIFAR100 (100 total classes) : 100 clients, each assigned a unique class
>
> This single-class assignment ensures maximal label distribution discrepancy between clients. Experiments were conducted with
> participation rate 40% and communication rounds 500.
> The results under this extreme setting are presented below.
> |  | | CIFAR10 | CIFAR100 |
> | ----- | ----- | ----- | -----  |
> | FedAvg | (AISTATS 2017) | 31.57±1.04 | 6.42±0.24 |
> | FedSAM |(ICML 2022) | 29.39±0.82 | 5.56±0.18 |
> | FedSMOO | (ICML 2023) | 10.00(failed) | 1.00(failed) |
> | FedLESAM-D |(ICML 2024) |  45.35±0.60 | 13.07±0.70 |
> | FedGMT  | (ours) | **58.23±0.68** | **22.13±0.34** |
> | FedGMTv2  |(ours) |  54.76±0.40 | 17.19±0.14 |
>
> The results shows that in an extreme non-IID federated learning setup where clients hold disjoint class partitions, FedGMT achieves state-of-the-art results: 58.23% on CIFAR10 and 22.13% on CIFAR100, outperforming FedLESAM-D (45.35%/13.07%) and other baselines. FedAvg (31.57%/6.42%) and FedSAM (29.39%/5.56%) struggle with label heterogeneity, while FedSMOO fails entirely. These results validate FedGMT’s superior robustness to extreme label skewness.
>
> ---
> **3.Comment on the challenge of privacy.** Our work builds upon the well-established FedAvg framework, inheriting its parameter-based communication architecture between server and clients. This design ensures full compatibility with existing privacy-preserving techniques developed for FedAvg, including mainstream defenses against privacy attacks such as differential privacy [A], homomorphic encryption [B], and secure multi-party computation [C].
>
> The core contributions of our paper focus on enhancing practical generalization accuracy for real-world datasets while minimizing communication and computation costs under data heterogeneity. Importantly, our methods impose no restrictions on the integration of privacy-preserving techniques, ensuring seamless compatibility with existing or emerging privacy frameworks.
>
> [A] Chen,H.,Vikalo,H.,etal. The best of both worlds: Accurate global and personalized models through federated learning with data-free hyper-knowledge distillation(ICLR , 2023).
>
> [B] Cai Y, Ding W, Xiao Y, et al. Secfed: A secure and efficient federated learning based on multi-key homomorphic encryption[J]. IEEE Transactions on Dependable and Secure Computing, 2023, 21(4): 3817-3833.
>
> [C] Chen L, Xiao D, Yu Z, et al. Secure and efficient federated learning via novel multi-party computation and compressed sensing[J]. Information Sciences, 2024, 667: 120481.
>
> ---
> It is a pleasure to discuss this with you, which will help us to further improve this work. Thank you again for reading this rebuttal.

---

> > ### Comment · Reviewer_5WUM · 2025-04-03
> >
> > I keep my (positive) rating :-)
> >
> > Just one comment on the non iid: I am well aware of the standard procedure in FedLM, but there of literature on learning in the context of drift or covariate shift which is not FL (an older highly cited one with reference to data sets https://www.sciencedirect.com/science/article/pii/S0925231217315928); I would appreciate to see broader challenges here as FL starts to overfit on the data --- not necessarily for this ICML paper but in the future....

---

> > > ### Author Response · Authors · 2025-04-04
> > >
> > > Thank you very much for your comment and affirmation of our work. We greatly appreciate your emphasis on the broader context of drift/covariate shift research, which aligns with our vision for advancing robust FL frameworks.
> > >
> > > We selected two drift datasets from the literature [A] you provided and one real - world scenario drift dataset to assess the performance of FL methods.  We will detail the dataset and implement in this part.
> > > - **Interchanging RBF**.  This dataset consists of fifteen Gaussians with random covariance matrices. Every 3000 samples, these Gaussians replace each other. With each replacement, the number of Gaussians changing their positions increases by one until all of them change their locations simultaneously. This setup enables us to evaluate an algorithm's performance in the face of abrupt drifts with increasing intensity. In total, there are 66 abrupt drifts within this dataset.
> > > - **Forest Cover Type**. This dataset assigns cartographic variables such as elevation, slope, and soil type of 30×30 - meter cells to different forest cover types. Only forests with minimal human - caused disturbances were considered, so the resulting forest cover types are mainly determined by ecological processes.
> > > - **HAR (Human Activity Recognition Using Smartphones DataSet).** [B] This dataset is constructed from the recordings of 30 subjects performing activities of daily living while carrying a waist - mounted smartphone with embedded inertial sensors. The data is naturally associated with each subject (client).
> > >
> > > For these three tabular datasets, we use a MLP with three hidden layers. The numbers of hidden units of three layers are 32, 16, and 8.  The remaining settings are consistent with those in our manuscript. The experimental results are presented below.
> > >
> > > |         |  | Inter. RBF | Cover type | HAR |
> > > | - | - | - | - |  - |
> > > | FedAvg |   |  15.85  ±  1.95  | 70.75±7.05      | 86.92±0.99         |
> > > | FedSAM |  | 16.14  ± 1.81   | 71.40±6.66       | 87.04±0.86         |
> > > | FedSMOO | |  17.57 ±   1.46   | 75.63±3.60       | 91.32±0.75  |
> > > | FedLESAM-D| |   **19.16 ±  1.47**   | 75.67±3.68      | 91.15±0.93  |
> > > | FedGMT|    | 18.56±1.05        | **78.27±0.97**       | **92.60±0.13** |
> > >
> > > FedGMT achieves the highest accuracy in the Forest Cover Type dataset (78.27 ± 0.97) and the HAR dataset (92.60 ± 0.13). This indicates that FedGMT is well - suited to handle the data characteristics and drifts present in these real - world datasets.
> > > Interchanging RBF dataset is designed to simulate abrupt drifts with increasing intensity. FedLESAM - D performs the best here with an accuracy of 19.16 ± 1.47. FedGMT follows closely with 18.56 ± 1.05. The relatively lower accuracy values for all methods on this dataset can be attributed to the complex and abrupt nature of the drifts. This shows that new methods are needed in FL to solve the drift of such datasets in the future.
> > >
> > > [A] Losing, Viktor, Barbara Hammer, and Heiko Wersing. "Incremental on-line learning: A review and comparison of state of the art algorithms." Neurocomputing 275 (2018): 1261-1274.
> > >
> > > [B] Anguita, Davide, et al. "A public domain dataset for human activity recognition using smartphones." Esann. Vol. 3. No. 1. 2013.
> > >
> > > ---
> > >
> > > Once again, thank you for your invaluable insights and suggestions. Your professional expertise and constructive suggestions will significantly improve the quality of our work.

---

### Official Review · Reviewer_aEZv · 2025-03-14

**Overall Recommendation:** 3

**Summary:**

This paper studies sharpness-aware minimization (SAM) in federated learning (FL) in the presence of data heterogeneity. The major problem for SAM in FL is that the clients cannot get an accurate estimate of the global objective/gradient due to heterogeneous data distribution. Existing literature has proposed utilizing the model updates in the previous communication round to approximate the global gradient. This paper proposes FedGMT, which leverages the global model trajectory which includes the pseudo-gradients from all the previous rounds through exponential moving averaging.

**Claims And Evidence:**

Yes.

**Essential References Not Discussed:**

The literature review looks comprehensive.

**Experimental Designs Or Analyses:**

Yes. In the experiments, it is mentioned that SGD with a learning rate of 0.01 is adopted. While the momentum of 0.9 is quite common and standard in the literature, I expect that the learning rate should be tuned from a reasonable set for the baselines for the sake of a fair comparison.

**Methods And Evaluation Criteria:**

Yes. Extensive experiments on both CV and NLP problems, with various datasets and neural network models are conducted.

**Other Comments Or Suggestions:**

The paper is well-written.

**Other Strengths And Weaknesses:**

**Strengths:**
1. The topic and problem that this paper aims to tackle are interesting and important.
2. The paper is well-written and not difficult to follow.
3. Extensive experiments on both CV and NLP tasks are conducted to validate the effectiveness of the proposed algorithms.

**Weakness:**
1. Incorporating KL divergence in training loss is not new. For example, the derived algorithm looks similar to the cited paper FedGKD [A], which also adopts the EMA at the server and adds a KL term to the local loss. Therefore, in my understanding, the major difference lies in the adaptation of ADMM, which was proposed in FedSMOO.
2. The improvement of the algorithm seems to arise mainly from ADMM since the performance of only using $L^{global}$ in the ablation study in Table 3 looks similar to Fed-SAM in Table 2. (78.18 vs. 78.31, 61.28 vs. 61.05 in Dir(1.0) and (0.01) conditions, respectively). More experimental results without ADMM would help demonstrate the effectiveness of the derived global sharpness measure via Global Model Trajectory, which is the major contribution of this paper based on my understanding.
3. The learning rates seem not carefully tuned for the baselines.

[A] Yao, D., Pan, W., Dai, Y., Wan, Y., Ding, X., Yu, C., ... & Sun, L. (2023). FedGKD: Toward heterogeneous federated learning via global knowledge distillation. IEEE Transactions on Computers, 73(1), 3-17.

**Questions For Authors:**

Please see my comments about weaknesses. In addition, could the authors help me understand how incorporating EMA and ADMM improves the learning rate for non-convex optimization from $O(1/\sqrt{T})$ to $O(1/T)$?

**Relation To Broader Scientific Literature:**

There are many existing works endeavoring to address the data heterogeneity and computational overhead issues in SAM for federated learning, such as FedSMOO, FedSpeed, and FedLESAM. This paper advances the literature by incorporating the global model trajectory.

**Theoretical Claims:**

Yes. I quickly went through the proofs (which seem correct), but have not had the time to check them step-by-step.

---

> ### Author Rebuttal · Authors · 2025-03-27
>
> We thank the reviewer for the positive review and constructive comments. We provide our responses as follows.
>
> ---
> **W1.FedGMT vs. FedGKD.** While FedGKD appears to be similar to our method, there are fundamental differences. Practically, FedGKD takes an element-wise average over the latest 5 rounds (as they suggest) of global models and sends it to clients. This averaging method isn't EMA; it demands extra storage space and doubles the communication overhead.
> We use EMA with KL divergence in the training loss because, as proven in Eq. (8), minimizing the loss difference $L(e)-L(w)$ between the EMA model $e$ and the global model $w$ is equivalent to minimizing the SAM’s sharpness measure for the global optimization. However, directly combining this loss difference with the FL objective will cancel out $L(w)$. So  we replace the cross entropy (CE) loss with the KL divergence loss to decouple the vanilla loss since minimizing the CE loss is equivalent to minimizing the KL loss. Thus, minimizing the KL loss in FedGMT aims to minimize global sharpness and reduce computational cost compared to other SAM-based methods, a goal that FedGKD cannot achieve.
> The comparative experiments in the following table in **W2** demonstrate that FedGMT outperforms FedGKD by 2.1% on Dir(0.1) and 2.79% on Dir(0.01) when both methods employ the ADMM.
>
> ---
> **W2. Methods comparison without ADMM.** We note that the performance of $L^{glotra}$  degrades without ADMM, which is due to non-vanishing biases between local update and global update in heterogeneity scenario. This aligns with methods such as FedSpeed, FedSMOO, and FedLESAM, which also incorporate ADMM to address such issues. Thus, approaches that omit ADMM may fail to fully leverage the proposed sharpness measure's potential. We evaluate performance with/without ADMM on CIFAR10 to demonstrate ADMM's necessity for maintaining optimization stability under data heterogeneity. The results are stated below.
> |  |  W/O ADMM Dir(0.1) | W/ ADMM Dir(0.1) |W/O ADMM Dir(0.01) | W/ ADMM Dir(0.01) |
> | -----| -----  | ----- | -----  |----- |
> | FedAvg|70.61±3.51|75.71±0.95 |61.94±4.93|70.52±2.32 |
> | FedSAM|70.96±3.97|77.51±0.97 |61.05±4.85|71.96 ±1.67 |
> | FedSMOO|71.28±3.90|77.08±0.97 |61.60±4.48|72.11 ±1.79|
> | FedLESAM|70.89±3.59|76.11±0.83|**62.95±4.60**|71.10 ±2.82|
> | FedGKD |72.54±2.68|77.07±0.83 |61.50±3.96|71.88±2.77|
> | FedGMT|**72.68±2.19**|**79.17±0.49**|61.28±3.11|**74.67 ±0.77**|
> | FedGMTv2 | 72.62±2.27|78.73±0.47|61.90±3.46|74.11±1.32|
>
> The above experimental results show that FedGMT with ADMM achieve best performance due to our sharpness measure can more accurate estimate global sharpness than others (Figure 2 also demonstrate this). We will add these experimental results in our updated version.
>
> ---
> **W3.Learning rate tuning.** Thanks for the suggestions. Learning rate which is a basic hyperparameters in the deep learning, and usually we do not finetune this for the fair comparison in the experiments. After a comprehensive review of existing federated learning literature, we observe that prior works consistently use a fixed learning rate across all compared methods. In our study, we  select learning rates from the set {0.001,0.01,0.1} for different datasets and model architectures to maximize SGD performance, then fix this optimal value for all methods. Specifically, a learning rate of 0.01 was employed for CIFAR10, CIFAR100, and CINIC10, while 0.1 was used for AG News.
>
> ---
> **Q1. Convergence rate understand.** In the federated stochastic non-convex setting, under standard assumptions of "smoothness," "bounded stochastic gradients," "heterogeneity," and other assumptions, prior works[A,B] have established $O(1/T)$ convergence rates for their algorithms. Specifically, our theoretical contributions lie in proving convergence without relying on the restrictive assumptions of bounded heterogeneity or requiring local minima in each communication round. This is enabled by our ADMM-based framework, which derives properties (e.g., Lemmas D.5 and D.6) to bound the global gradient norm in Eq. (34).
> Prior work [C] establishes faster convergence under the assumption of client similarity in non-convex settings.  Our Eq. (10) scales $L^{glotra}$ as a non-negative convex function which can be seen as an exponential moving average of historical global gradients into local updates across all clients. This shared similarity in EMA across clients reduces variance between local and global updates, thereby accelerating convergence.
>
> [A] Fedpd: A federated learning framework with adaptivity to non-iid data. IEEE Transactions on Signal Processing, 69:6055–6070, 2021.
>
> [B] Fedadmm: A federated primal-dual algorithm allowing partial participation. arXiv preprint arXiv:2203.15104, 2022.
>
> [C] Scaffold:Stochastic controlled averaging for federated learning(ICML 2020).
>
> ---
> It is a pleasure to discuss this with you, which will help us to further improve this work. Thank you again for reading this rebuttal.

---

> > ### Comment · Reviewer_aEZv · 2025-04-03
> >
> > Thanks for the efforts addressing my comments. I have some additional comments are as follows.
> >
> > 1. Since this paper is concerned with a new optimization method, a careful tuning may be preferred for a fair performance comparison (considering that SGD is sensitive to learning rate).
> > 2. In general, the best rate a first-order method can attain is $O(1/\sqrt{T})$ for non-convex optimization. I was curious which part of the proposed algorithm boosts the convergence rate to $O(1/T)$.

---

> > > ### Author Response · Authors · 2025-04-04
> > >
> > > Thanks for your valuable comments to our work.  We are happy to continue the discussion with you. We are also very honored to share some of our understandings with you.
> > >
> > > ---
> > > **1. Experiments with learning rate tuning.** We conducted experiments on CIFAR10-Dir(0.1) for each method with local different learning rates from  {0.001,0.01,0.1}. For our FedGMT, we fix the learning rate decay at 0.998.  The remaining hyperparameters are the same as those described in Section 4.1 (FedGMT Setting). For other methods, we refer to their official  papers to search the best hyperparameters for their performance. These hyperparameters mainly include the SAM perturbation rate {0.001, 0.01 , 0.1} , penalized coefficient {1, 10, 100} and learning rate decay {0.998,0.9998,0.99998}. The results are stated below.
> > >
> > > |         |  | lr = 0.001 | lr = 0.01 | lr =0.1 |
> > > | - | - | - | - |  - |
> > > | FedAvg |  (AISTATS 2017) | 54.61±5.00       | 71.70±3.30       | 71.76±1.88         |
> > > | FedSAM |  (ICML 2022) | 54.63±4.99      | 71.73±3.22       | 72.50±1.95         |
> > > | FedSMOO | (ICML 2023) | 70.94±1.22        | 77.49±0.57       | 77.48±1.05  |
> > > | FedLESAM-D| (ICML 2024) | 70.80±1.20        | 75.99±0.96       | 75.71±0.79  |
> > > | FedGMT|  (Ours)  | **74.85±0.63**        | **79.84±0.27**       | **78.21±0.55** |
> > >
> > > The result show that FedGMT still achieves superior accuracy than other baselines after a careful tuning. In addition, we find the SAM perturbation need to be carefully tuned for different local learning rate. In practice, FedGMT ​​can achieve higher performance compared to other baselines without the complex tuning.
> > >
> > > ---
> > >
> > > **2. Convergence rate explanation.**
> > > The essence of this problem is to understand the role of local steps (denoted as $K$) in FL algorithms. To better understand this issue, we first review some foundational theories.
> > >
> > > On the one hand, non-ADMM FL algorithms aim to solve $min L(w) = \sum L_m(w)$. In this problem, prior studies typically achieve a convergence rate of $O(1/ \sqrt {T})$ under non-convex condition. For example, [A] demonstrates that both FedAvg and SCAFFOLD attain $O(1/ \sqrt {TK})$. Here, the K is regarded as a constant and in such cases, these algorithme are $O(1/ \sqrt {T})$ rate.
> > >
> > > On the other hand, the problem of ADMM-based FL algorithms aim to solve $min L(w) = \sum L_m(w_m)\ \ s.t. w_m = w$. Note that when this problem finally converges to satisfy the constraint, these two problems are equivalent. By constructing the Lagrangian relaxation (Eq. (11)), the constrained problem is transformed into unconstrained form. Each client solves a local subproblem per round (Eq. (12)), and convergence analysis assumes clients approximate stationary points for these subproblems. Under this assumption, our proposed algorithm achieve $O(1/ T)$ convergence rates independent of local steps $K$. However, in practice we use K local steps to solve each sub-problem. This means $K$ needs to be long enough to approximate the solution of Eq. (12).  Actually, [B] prove that $K$ need to satisfy $K = O(T)$ in ADMM-based FL algorithms to achieve $O(1/ T)$.
> > > Substituting $K = O(T)$ into non-ADMM methods' $O(1/ \sqrt {TK})$ rates (e.g., FedAvg/SCAFFOLD [A]) also yields $O(1/ T)$.
> > >  **So this means that their convergence rate is actually equivalent. The difference between them is that ADMM-based algorithms need less comunication rounds.** For example, if we need to compute $T$ times gradients to satisfy the target acc or error, theoretically, FedAvg and SCAFFOLD need to comunicate $T/K$ times, but ADMM-based algorithms need $\sqrt T$ times comunication with $K = O(T)$. This theoretical advantage aligns with empirical results (Figure 5), where ADMM-based algorithms converge faster.
> > > However, enforcing $K = O(T)$ in FedAvg or SCAFFOLD introduces polynomial divergence in their proofs. So we usually regard $K$ as a constant in these non-ADMM based algorithms.
> > >
> > > In summary, our algorithm boosts the convergence rate to $O(1/ T)$ holds under the condition of local client approach a stationary point or $K = O(T)$.  This is not a violation the best rate a first-order method can attain is $O(1/ \sqrt {T})$ for non-convex optimization since the $O(1/ T)$ rate relies on additional assumptions beyond standard non-convex settings.
> > >
> > > [A] Scaffold:Stochastic controlled averaging for federated learning(ICML 2020).
> > >
> > > [B] Fedspeed:Larger local interval,less communication round, and higher generalization accuracy (ICLR 2023).
> > >
> > > ---
> > > Thank you again for reading our rebuttal. We have tried to address most if not all concerns raised by the reviewers. If our responses have sufficiently addressed your questions/concerns, it will be great if you are willing to kindly improve your score. Thank you.

---

### Decision · Program_Chairs · 2025-05-01

**Decision:**

Accept (poster)

**Comment:**

This works tackles sharpness‑aware minimization (SAM) in the federated learning setting by directly measuring global sharpness via a global model trajectory, rather than per‑client approximations. Its key ingredients are: (1) The server maintains an exponential moving average (EMA) of past global models. (2) Clients add a KL‑divergence term between their local model’s outputs and the EMA model’s outputs, which provably aligns with minimizing the global SAM sharpness measure. (3) A dual variable per client is used to enforce consistency between local and global updates. Further, this approach avoids the second backward pass of standard SAM, cutting compute by roughly half. This work also conducted some extensive experiments to justify their approach.  The reviewers all recognize the positive contribution of this work and the capability of the proposed approach to trade-off accuracy and efficiency tradeoff. Therefore I recommend acceptance of the paper.